# Cyanobacteria net community production in the Baltic Sea as inferred from profiling $p\mathrm{CO_2}$ measurements

Jens Daniel Müller[1,2], Bernd Schneider[1], Ulf Gräwe[3], Peer Fietzek[4], Marcus Bo Wallin[5,6], Anna Rutgersson[5], Norbert Wasmund[7], Siegfried Krüger[3], and Gregor Rehder[1]

[1]Department of Marine Chemistry, Leibniz Institute for Baltic Sea Research Warnemünde, Rostock, Germany
[2]Environmental Physics, Institute of Biogeochemistry and Pollutant Dynamics, ETH Zurich, Zurich, Switzerland
[3]Department of Physical Oceanography and Instrumentation, Leibniz Institute for Baltic Sea Research Warnemünde, Rostock, Germany
[4]Kongsberg Maritime Germany GmbH, Hamburg, Germany
[5]Department of Earth Sciences, Uppsala University, Uppsala, Sweden
[6]Department of Aquatic Sciences and Assessment, Swedish University of Agricultural Sciences, Uppsala, Sweden
[7]Department of Biological Oceanography, Leibniz Institute for Baltic Sea Research Warnemünde, Rostock, Germany

**Correspondence:** Jens Daniel Müller (jensdaniel.mueller@usys.ethz.ch)

**Abstract.**

Organic matter production by cyanobacteria blooms is a major environmental concern for the Baltic Sea, as it promotes the spread of anoxic zones. Partial pressure of carbon dioxide ($p\mathrm{CO_2}$) measurements carried out on Ships of Opportunity (SOOP) since 2003 have proven to be a powerful tool to resolve the carbon dynamics of the blooms in space and time. However, SOOP

measurements lack the possibility to directly constrain depth–integrated net community production (NCP) in moles of carbon per surface area due to their restriction to the sea surface. This study tackles the knowledge gap through (1) providing an NCP best–guess for an individual cyanobacteria bloom based on repeated profiling measurements of $p\mathrm{CO_2}$ and (2) establishing an algorithm to accurately reconstruct depth–integrated NCP from surface $p\mathrm{CO_2}$ observations in combination with modelled temperature profiles.

Goal (1) was achieved by deploying state–of–the–art sensor technology from a small–scale sailing vessel. The low–cost and flexible platform enabled observations covering an entire bloom event that occurred in July – August 2018 in the Eastern Gotland Sea. For the biogeochemical interpretation, recorded $p\mathrm{CO_2}$ profiles were converted to $\mathrm{C_T}$*, which is the dissolved inorganic carbon concentration normalised to alkalinity. We found that the investigated bloom event was dominated by *Nodularia* and had many biogeochemical characteristics in common with blooms in previous years. In particular, it lasted for about

three weeks, caused a $\mathrm{C_T}$* drawdown of 90 $\mu$mol kg$^{-1}$, and was accompanied by a sea surface temperature increase of 10 °C. The novel finding of this study is the vertical extension of the $\mathrm{C_T}$* drawdown up to the compensation depth located at around 12 m. Integration of the $\mathrm{C_T}$* drawdown across this depth and correction for vertical fluxes leads to an NCP best–guess of ~1.2 mol–C m$^{-2}$ over the productive period.

Addressing goal (2), we combined modelled hydrographical profiles with surface $p\mathrm{CO_2}$ observations recorded by SOOP

*Finnmaid* within the study area. Introducing the temperature penetration depth (TPD) as a new parameter to integrate SOOP

observations across depth, we achieve an NCP reconstruction that agrees to the best–guess within 10%, which is considerably better than the reconstruction based on a classical mixed layer depth constraint.

Applying the TPD approach to almost two decades of surface $p\mathrm{CO_2}$ observations available for the Baltic Sea bears the potential to provide new insights into the control and long–term trends of cyanobacteria NCP. This understanding is key for an effective design and monitoring of conservation measures aiming at a Good Environmental Status of the Baltic Sea.

# 1 Introduction

## 1.1 Net community production (NCP) in marine ecosystems

Net community production (NCP) of organic matter triggers many biogeochemical processes that control the functioning and state of marine ecosystems. Globally relevant examples are the biological carbon pump (Henson et al., 2011; Sanders et al., 2014) and the establishment of oxygen minimum zones (Gilly et al., 2013; Oschlies et al., 2018). In this biogeochemical context, we define NCP as the net amount of carbon fixed in organic matter (gross production minus respiration) that is produced in a defined water volume over a defined period. This definition implies that the choice of an integration depth is a critical component of any NCP estimate. Traditionally, NCP is constrained either to the depth of the euphotic zone, the compensation depth at which gross production equals respiration, or the mixed layer depth (Sarmiento and Gruber, 2006). Of those approaches, only the integration to the compensation depth is directly linked to the vertical distribution of carbon fixation and remineralisation and therefore quantifies the amount of formed organic matter that can potentially be exported. The reliable quantification of this potential export is a prerequisite to understand subsequent biogeochemical transformation of the organic matter and its imprint on environmental conditions in any aquatic system.

## 1.2 Baltic Sea

On a regional scale, NCP quantification is of particular importance to study deoxygenation of stratified water bodies caused by the remineralisation of organic matter that was exported across a permanent pycnocline. This hydrographical situation is typically encountered in semi–enclosed, silled estuaries such as the Baltic Sea. The deep basins of the Baltic Sea receive substantial amounts of oxygenated, salty water from the North Sea only during occasional major inflow events. Between inflow events, those water masses can stagnate for more than a decade below the permanent halocline (Mohrholz et al., 2015), which is located at around 60 m water depth in the Central Baltic Sea. The export of organic matter into the deep waters is considered the ultimate cause for the expansion of anoxic areas in the Baltic Sea, which are nowadays among the largest anthropogenically induced anoxic areas in the world (Carstensen et al., 2014). Although the actual oxygenation state of the deep basins of the Baltic Sea is modulated by the frequency and strength of inflow events (Mohrholz et al., 2015; Neumann et al., 2017) and the biogeochemical properties of the inflowing waters (Meier et al., 2018), the long–term expansion of the anoxic water body was primarily attributed to increased nutrient inputs from land (Jokinen et al., 2018; Meier et al., 2019; Carstensen et al., 2014; Mohrholz, 2018) that fueled the organic matter production in surface waters. Therefore, a quantitative and mechanistic

understanding of organic matter production is key to understand, predict, and eventually counteract the expansion of the anoxic areas. Such measures to reduce eutrophication and deep water anoxia actually represent a core component of the EU Marine Strategy Framework Directive (MSFD), which is implemented as the HELCOM Baltic Sea Action Plan (BSAP) and aims at a Good Environmental Status (GES) of the Baltic Sea.

## 1.3 Cyanobacteria blooms

The annual cycle of organic matter production in the Central Baltic Sea can be broadly divided into two phases (Schneider and Müller, 2018). The first production phase is the spring bloom, which is controlled by the availability of nitrate and shifted from being dominated by diatoms to dinoflagellates in the late 1980s (Wasmund et al., 2017; Spilling et al., 2018). After a so-called blue water period with close–to–zero NCP rates, the second production phase consists of mid–summer blooms dominated by nitrogen-fixing cyanobacteria that develop in most years depending on meteorological conditions. Although cyanobacteria NCP is yet poorly constrained, its relative contribution to the annual NCP in the Eastern Gotland Sea in 2009 was estimated in the order of 40% (Schneider and Müller, 2018; Schneider et al., 2014), though the uncertainty of this estimate is high. This preliminary estimate further needs to be interpreted with care as cyanobacteria NCP varies significantly between years and regions. The blooming of cyanobacteria is limited to the months of June to August (Kownacka et al., 2020) and represents a common feature of the Baltic Sea ecosystem at least since the 1960s (Finni et al., 2001). The blooms are a major public concern, because they produce toxins and form thick surface scums lowering the recreational value of the Baltic Sea. From a biogeochemical perspective, the ability to fix nitrogen makes cyanobacteria independent from nitrate and aggravates the eutrophication state of the Baltic Sea. Whether their growth is limited by the availability of phosphate remains an ongoing debate (Nausch et al., 2012), although the highly variable C:P ratio of their biomass (Nausch et al., 2009) indicates phenotypic plasticity. Other ongoing debates in the field of cyanobacteria research address the fate of the produced organic matter and its transfer into the food web (Karlson et al., 2015), the intensification of cyanobacteria blooms through positive feedback loops between organic matter production, deep water anoxia and the release of phosphate from anoxic sediments (Vahtera et al., 2007), as well as their response to ongoing changes in salinity, temperature and the partial pressure of carbon dioxide, $p\text{CO}_2$ (Olofsson et al., 2019, 2020). The limited understanding of the factors that control the blooms hinders the reliable prediction of the future state of the Baltic Sea and therefore the prioritisation of conservation measures (Elmgren, 2001). In particular, it remains challenging to disentangle how expected trends – including warming, reduced nutrient loads, and increasing $p\text{CO}_2$ – might impact cyanobacteria growth (Meier et al., 2019; Saraiva et al., 2019). A long–term hindcast of cyanobacteria NCP and the attribution of its strength to prevailing environmental conditions in particular years could improve our understanding of controlling factors and facilitate more reliable predictions of the blooms. However, such a hindcast of cyanobacteria NCP was so far impossible due to missing vertically-resolved observations that would allow to constrain their organic matter production.

## 1.4 Quantification of NCP

Striving for a better understanding of the ecosystem impact of cyanobacteria blooms, the accurate quantification of produced organic matter is key. In this regard, NCP could in principle be quantified directly as an increase in particulate organic carbon

(POC). However, POC measurements would not detect the amount of organic matter that was exported between observations (Wasmund et al., 2005) and also fail to achieve the required spatio–temporal resolution due to a low degree of automation. As an alternative, it is possible to quantify NCP through the drawdown of dissolved inorganic carbon ($C_T$) from the water column (Schneider et al., 2003). From a biogeochemical perspective, the determination of NCP in terms of carbon is ideal, because carbon is the major component of organic matter and directly related to the amount of oxygen ($O_2$) that is consumed

during remineralisation. In principle, NCP could as well be estimated from $O_2$ time series. However, the equilibrium reactions of carbon dioxide ($CO_2$) in seawater result in slower re–equilibration of $CO_2$ with the atmosphere compared to $O_2$ (Wanninkhof, 2014). This results in substantially longer preservation of the $C_T$ signal and thus a lower uncertainty contribution of required air–sea $CO_2$ flux corrections, making $C_T$ the preferred tracer for NCP. During the Baltic Sea spring bloom, the tracing of nutrient drawdown is a meaningful alternative to quantify NCP and convincingly leads to comparable results to the

$C_T$ approach (Wasmund et al., 2005). However, time series of nutrient drawdown do not allow for determining NCP of algae blooms dominated by nitrogen-fixing organisms and those with highly variable C:P ratios. As both characteristics are typical for Baltic Sea cyanobacteria blooms (Nausch et al., 2009), the well established $C_T$ approach remains the favorable method to determine mid–summer NCP in this region. However, it should be noted that NCP estimates derived from this approach include the formation of POC and dissolved organic carbon (DOC). The produced DOC contributes ~20% to NCP (Hansell

and Carlson, 1998; Schneider and Kuss, 2004) and is not likely to be vertically exported.

### 1.5 Previous studies

Among previous attempts to trace and quantify the organic matter production of cyanobacteria blooms, automated $pCO_2$ measurements on the Ship of Opportunity (SOOP) *Finnmaid* played a pivotal role. Those measurements were started in 2003 and it was demonstrated that highly accurate time series of changes (not absolute values) in $C_T$ can be derived from $pCO_2$ obser-

105 vations (Schneider et al., 2006). The conversion from $pCO_2$ to $C_T$ relies on a fixed alkalinity ($A_T$) estimate and is applicable under the condition that internal sinks and sources of $A_T$ can be excluded, which is the case in the Baltic Sea due to the absence of calcifying plankton (Tyrrell et al., 2008). The derived parameter is comparable to directly measured $C_T$ normalised to $A_T$, and in the following referred to as $C_T$*. For several years of SOOP observations, it was shown that the $C_T$* drawdown during mid–summer cyanobacteria blooms occurs in pulses of days to weeks, primarily during calm, sunny days. Further, it

was found that the $C_T$* drawdown correlates well with the co–occurring increase in sea surface temperature (SST), rather than with absolute SST. This relationship was attributed to a common driver, which is the light dose received by the water mass under consideration (Schneider and Müller, 2018).

Despite the successful investigation of cyanobacteria blooms through SOOP $pCO_2$ observations, providing a depth–integrated estimate of NCP in units of moles carbon fixed per surface area remains challenging due to the restriction of SOOP observations

to surface waters. Previous studies aiming at a depth–integrated NCP estimate either simply assumed that the $C_T$* drawdown reached as far down as the water inlet of the measurement system (Schneider and Müller, 2018) or relied on a modelled mixed layer depth for the vertical integration of surface observations (Schneider et al., 2014). However, in the absence of any vertically resolved measurements, neither approach could be validated. Likewise, remote sensing approaches were capable to resolve the

spatial coverage of the blooms (Hansson and Hakansson, 2007; Kahru and Elmgren, 2014), but failed to detect their vertical extent (Kutser et al., 2008) and quantify NCP. Finally, regular research vessel cruises allowed for the determination of a full suite of biogeochemical parameters from discrete water samples and even the experimental determination of carbon fixation rates through [14]C incubations (Wasmund et al., 2001, 2005). Such incubation experiments can provide valuable information about instantaneous rates of NCP, but – in contrast to time series observations such as obtained by SOOP measurements – do not allow to integrate observed changes over time and constrain budgets of biogeochemical transformations. This integration over time requires several weeks of repeated observations to resolve the progression of entire bloom events, ideally covering a station network to average bloom patchiness.

## 1.6 This study

This study builds upon the previous success to determine NCP based on $pCO_2$ time series, but extends the approach to vertically resolved observations for the first time. The primary goals of this study are to

(1) provide a best–guess for the depth–integrated NCP of an individual cyanobacteria bloom based on the full suite of depth–resolved in situ measurements and

(2) establish an algorithm to reconstruct depth–integrated NCP based on surface $pCO_2$ observations and modelled hydrographical profiles

Achieving goal (2) and applying the algorithm to almost two decades of SOOP $pCO_2$ observations in the Baltic Sea would not only allow to determine long–term trends of cyanobacteria NCP, but also enable disentangling its drivers through a comparison of NCP estimates from different years characterized by particular environmental conditions such as SST, $pCO_2$ and nutrient availability.

## 2 Methods

### 2.1 Overview

Profiling in situ sensor measurements and water sampling were performed on board the 27ft sailing vessel SV *Tina V* in the framework of the field sampling campaign "BloomSail". The study area was located in the Central Baltic Sea and extended about 25 nautical miles from the coast of Gotland into the Eastern Gotland Basin (Fig. 1). Measurements were performed during eight cruises covering the period July 6 to August 16, 2018 (Fig. 2).

A custom–made sensor package configured at IOW's Innovative Instrumentation department was deployed to perform $pCO_2$ and conductivity, temperature and depth (CTD) measurements. The sensor package was either towed near the sea surface while cruising or lowered to at least 25 m water depth at designated profiling stations. This study focuses exclusively on the vertical profiles recorded at stations 02 – 12 (Fig. 1b), whereas profiles at stations with water depths less than 60 m were not taken into account to avoid the impact of coastal processes. In addition to the sensor measurements, discrete samples for dissolved

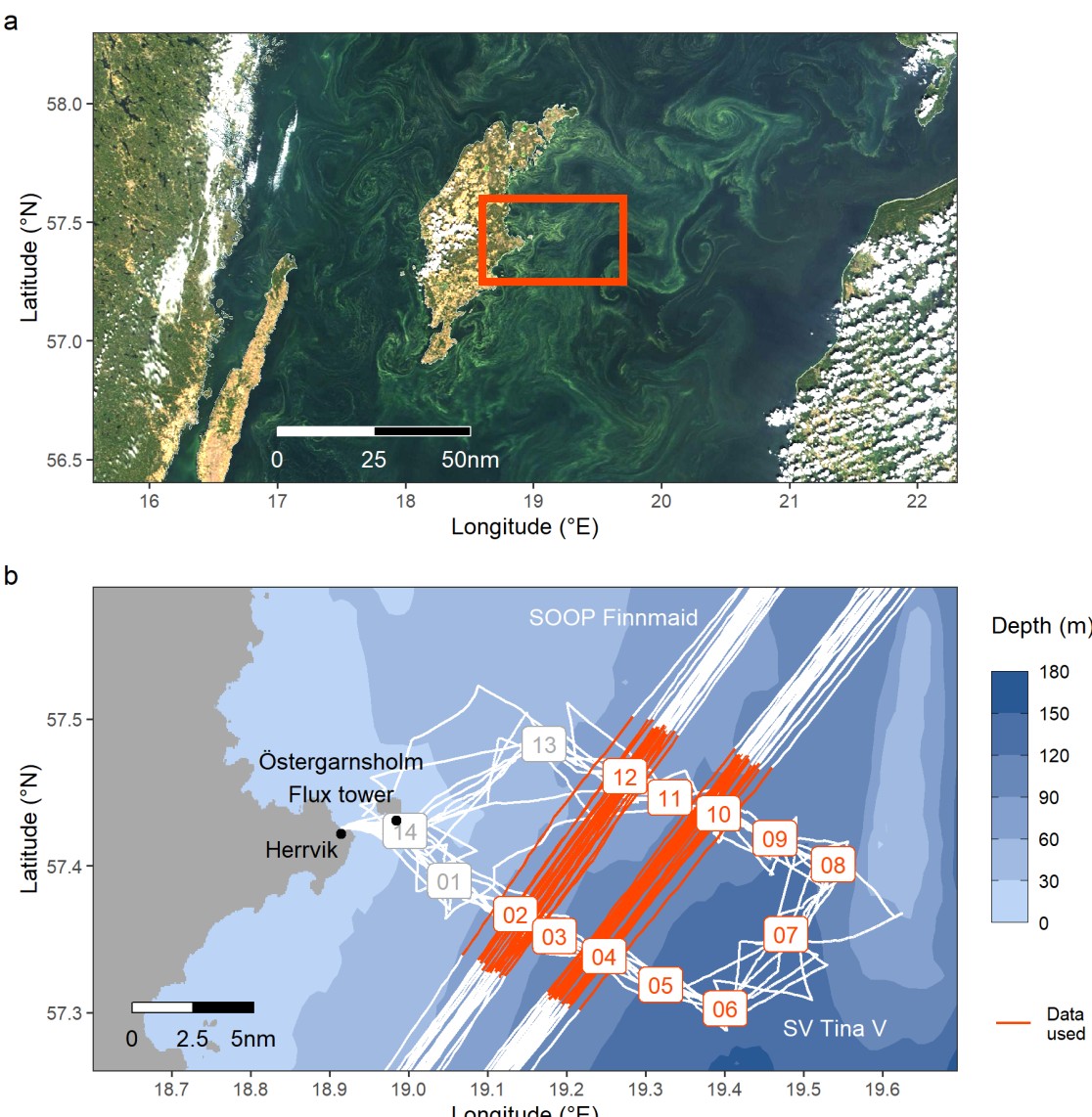

**Figure 1.** (a) Extent of the cyanobacteria bloom on July 26, detectable as greenish patterns in a true color satellite image (MODIS Aqua/Terra, Nasa Worldview) showing the Central Baltic Sea around the island of Gotland. The box indicates the study area as shown in (b), a bathymetric map with the cruise tracks of SV *Tina V* (BloomSail campaign) and SOOP *Finnmaid*. BloomSail stations and the SOOP sub–transects used in this study are highlighted in red. The ICOS flux tower for atmospheric measurements is located on the island of Östergarnsholm.

inorganic carbon ($C_T$), total alkalinity ($A_T$) and phytoplankton counts were collected. Track coordinates were continuously
recorded with a tablet computer (Galaxy Tab Active, Samsung Electronics, Suwon, South Korea).

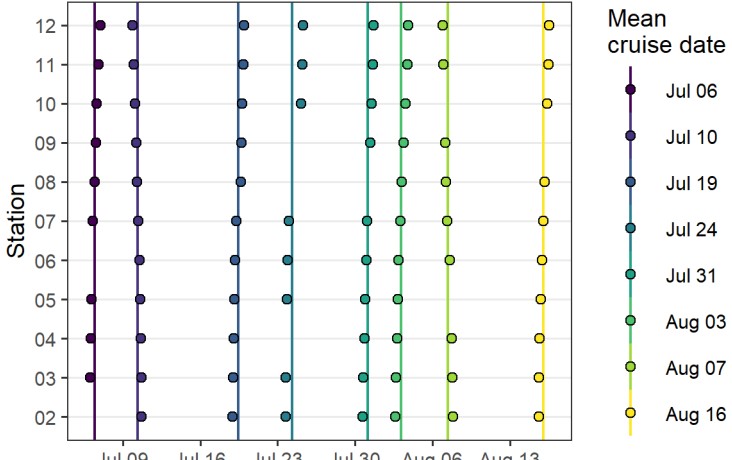

**Figure 2.** Overview on profiling sensor measurements performed at stations 02 – 12 (Fig. 1). Individual sampling events are displayed as points, whereas vertical lines indicate the mean date of each cruise event.

In addition to the field sampling campaign, atmospheric measurements of wind speed and $pCO_{2,atm}$ were provided by an ICOS (Integrated Carbon Observation System) station permanently operated on the island Östergarnsholm (Fig. 1b). Furthermore, sea surface $pCO_2$ and temperature (SST) were also determined on the SOOP *Finnmaid*, regularly crossing the field study area (Fig. 1b). High–resolution hydrographical model data were obtained from the Generalized Estuarine Turbulence Model (GETM) along a vertical section following the *Finnmaid* track.

## 2.2    Field sampling campaign

### 2.2.1    CTD measurements

CTD measurements were performed with a SBE 16 SEACAT instrument (serial number 2557; Sea-Bird Electronics, Bellevue, USA). Pre- and post-deployment calibrations of the instrument were carried out in the accredited calibration laboratory of the IOW in the time span of a few month around the deployments and confirmed that the temperature and conductivity sensors achieved the typical accuracy of better than $\pm0.01$ °C and $\pm0.01$ S m$^{-1}$, respectively. The manual operation of the sensor package was guided by real–time display of data submitted through a strain–relieved cable. Data stored on an internal memory were used for analysis. The CTD logging frequency was 15 seconds and observations were linearly interpolated to match the higher measurement frequency of the $pCO_2$ sensor (for additional details see Appendix A2). The CTD instrument supplied auxiliary sensors with power and served as a central unit to record and transmit analogue output signals.

### 2.2.2 $p$CO$_2$ sensor measurements

The submersible CO$_2$ sensor used in this study, a CONTROS HydroC® CO$_2$ (formerly Kongsberg Maritime Contros, Kiel, Germany; now -4H–JENA engineering, Jena, Germany), uses membrane equilibration of a headspace and subsequent optical Non–Dispersive Infra–Red (NDIR) absorption to determine the $p$CO$_2$ in water (Fietzek et al., 2014).

A pre– and post–deployment calibration of the sensor was performed by the manufacturer. $p$CO$_2$ data were post–processed taking into account the pre– and post–deployment calibration polynomials, as well as zeroing signals regularly recorded during each deployment. Given the statistics of the pre– and post–deployment calibration, the small drift encountered throughout the deployment and the otherwise smooth performance and regular cleaning of the sensor during the deployment, the accuracy of the drift corrected $p$CO$_2$ data is considered to be within 1% of reading as also found by Fietzek et al. (2014). For details

concerning sensor calibration, configuration, and signal post–processing, see Appendices A1 – A3.

Although the $p$CO$_2$ sensor achieves low and reproducible response times through active pumping of water onto the membrane, a correction of the response time ($\tau$) was applied following previously developed procedures (Miloshevich et al., 2004; Fiedler et al., 2013; Atamanchuk et al., 2015). After the response time correction, the mean absolute $p$CO$_2$ difference between the up– and downcast profile was <2.5 $\mu$atm in the upper 5 m of the water column and <7.5 $\mu$atm across the upper 20 m (Fig.

A2). For details concerning the response time correction, see Appendix A4.

The biogeochemical interpretation of the $p$CO$_2$ data was based on downcast profiles only. Since downcasts were started after complete equilibration of the $p$CO$_2$ sensor in near–surface waters, the applied response time correction has only a minor impact on the derived NCP estimate.

### 2.2.3 Discrete C$_T$, A$_T$ and phytoplankton sampling

Discrete water samples were collected with a manually released Niskin bottle. The sampling was restricted to stations 07 and 10 (Fig. 1b) due to logistic constraints. The sampling depth was estimated based on the length of the released line. C$_T$ and A$_T$ samples were filled into 250 ml SCHOTT–DURAN bottles and poisoned with 200 $\mu$L saturated HgCl$_2$ solution within 24 hours after sampling. Samples were stored dark and cool, transported to IOW, and analysed in the laboratory within no more than 21 days after sampling. C$_T$ was determined with an Automated Infra Red Inorganic Carbon Analyzer (AIRICA, MARIANDA,

Kiel, Germany) and A$_T$ was analysed by open cell titration (Dickson et al., 2007). C$_T$ and A$_T$ measurements were referenced to certified reference materials from batch 173 (Dickson et al., 2003). The mean observed A$_T$ was used for the calculation of C$_T$* from $p$CO$_2$ (see Sect. 2.5.2), while measured C$_T$ was only used for comparison to calculated values and not directly included in the NCP calculation. Phytoplankton samples were fixed with Lugol solution, and cyanobacteria community composition and biomass were determined by microscopic counts of the genera *Aphanizomenon*, *Dolichospermum* and *Nodularia* according to

the Utermöhl method (HELCOM, 2017). For details on the analysis of discrete samples, see Appendix B.

## 2.3 Atmospheric measurements

Meteorological observations were provided by the ICOS flux tower (Fig. 1b) located on the southernmost tip of the Island of Östergarnsholm (57.43010 °N, 18.98415 °E; Rutgersson et al., 2020). Atmospheric $pCO_{2,atm}$ was recorded with an atmospheric profile system (AP200, Campbell Scientific, Logan, USA) mounted with a $CO_2/H_2O$ gas analyzer (LI-840A, LI-COR Biosciences, Lincoln, USA). Wind speed was measured with a wind monitor (Young, Michigan, USA) at 12 m above mean sea level. Wind speed and $pCO_2$ data were averaged over 30 min intervals for further analysis. Measured wind speed was converted to $U_{10}$, the wind speed at 10 m above sea level (Winslow et al., 2016), to be consistent with the gas exchange parameterisation (see Sect. 2.5.2).

## 2.4 $C_T$* calculation

The dissolved inorganic carbon concentration ($C_T$*) was calculated from the measured profiles of temperature and response time corrected $pCO_2$ (Schneider et al., 2014), as well as the mean $A_T$ (1720 $\mu$mol kg$^{-1}$) and mean salinity (6.9) determined from discrete samples collected across the upper 20 m of the water column and over the entire observation period (Fig. B1). Calculations were performed with the R package seacarb (Gattuso et al., 2020), using the $CO_2$ dissociation constants for estuarine waters from Millero (2010).

The calculated $C_T$* represents an alkalinity– and salinity–normalised estimate of the dissolved inorganic carbon concentration. $C_T$* is suitable to accurately determine changes rather than absolute values of the dissolved inorganic carbon concentration and therefore the preferred variable to quantify NCP. The uncertainty in the determination of changes of $C_T$* is below 2 $\mu$mol kg$^{-1}$ when the mean $A_T$ is constrained within the observed standard deviation of $\pm 27$ $\mu$mol kg$^{-1}$ (see Appendix C1 for a detailed assessment).

## 2.5 NCP best–guess

The determination of our NCP best–guess relies on the interpretation of observed temporal changes in $C_T$* ($\Delta C_T$*) across the water column. Conceptually, our calculations follow the idea of a one–dimensional box model approach, which does not resolve regional variability within the research area, i.e. it neglects lateral water mass transport. The calculation of the underlying $\Delta C_T$* profiles requires a vertical gridding of measured profiles into discrete depth intervals $\delta z$ and their regional averaging across all stations (for details see Sect. 2.5.1). According to equation 1, we derive the column inventory of incremental changes of $\Delta C_T$* ($i\Delta C_T$*) between two cruise events through vertical integration of $\Delta C_T$* from the sea surface to the compensation depth (CD), i.e. the depth (z) at which no net drawdown of $CO_2$ was observed:

$$i\Delta C_T^* = \sum_{z=0m}^{CD} \Delta C_T^*(z)\, \delta z \tag{1}$$

Correcting $i\Delta C_T$* for the cumulative $CO_2$ fluxes between two cruise events caused by air–sea gas exchange ($F_{air-sea}$, see Sect. 2.5.2) and vertical mixing ($F_{mix}$, see Sect. 2.5.3) leads to incremental NCP estimates according to:

$$NCP_{best-guess} = -i\Delta C_T^* - F_{air-sea} - F_{mix} \tag{2}$$

Incremental NCP estimates between cruise events are further added up to derive cumulative NCP over the study period. We refer to the derived NCP estimate as our best–guess, as it is well-constrained by high-quality measurements and therefore as close to the truth as currently possible.

### 2.5.1 Vertical gridding and regional averaging

The vertical gridding of individual profiles was achieved by calculating mean values within depth intervals ($\delta z$) of 1m. Downcast profiles with missing observations from two or more depth intervals caused by zeroing measurements of the $pCO_2$ sensor were discarded, which affected 8 out of 86 recorded profiles. For each of eight cruise events (Fig. 2), regionally averaged profiles were further calculated as mean values within each depth interval across all stations. Based on those mean, vertically gridded cruise profiles, incremental and cumulative changes over time were calculated for each depth interval. Throughout the manuscript, observations averaged across the upper 0 – 6 m of the water column are referred to as surface observations.

### 2.5.2 Air–sea CO$_2$ flux

The air–sea gas exchange of $CO_2$ ($F_{air-sea}$) was calculated from sea surface $pCO_2$, salinity and temperature, in combination with $pCO_{2,atm}$ and $U_{10}$ according to Wanninkhof (2014). For the calculation, sea surface observations were linearly interpolated to match the temporal resolution of atmospheric measurements. A negative sign of $F_{air-sea}$ indicates uptake of $CO_2$ from the atmosphere.

### 2.5.3 Vertical entrainment flux of CO$_2$ through mixing

Between June 6 and August 7, vertical mixing of $C_T$* into the surface layer ($F_{mix}$) was neglected, because a stable thermocline coincided with the integration depth for the NCP calculation (i.e. the compensation depth). However, clear signals for significant vertical entrainment of $C_T$* across this layer were observed between August 7 and 16 (Fig. 3). This entrainment was quantified assuming an instantaneous complete vertical mixing to 17 m water depth after August 7. For this simplified scenario, $F_{mix}$ was estimated based on a mass–balance of $C_T$*, which behaves conservatively with respect to mixing (see Appendix C2 for details). A negative sign of $F_{mix}$ indicates entrainment of $CO_2$ into the surface layer.

### 2.6 NCP reconstruction from surface $p$CO$_2$ observations and hydrographical profiles

Our calculation of depth–integrated NCP from a time series of surface $pCO_2$ observations, such as provided by SOOP lines, also relies on the conversion of $pCO_2$ to $C_T$*. Incremental changes of $C_T$* in the surface water further need to be multiplied with an estimate of the integration depth (ID) to derive an inventory change. Taking the air–sea flux of $CO_2$ (Sect. 2.5.2) into account, the NCP reconstruction can be determined as:

$$NCP_{reconstruction} = -\Delta C_T*_{surface} \cdot ID - F_{air-sea} \tag{3}$$

In the lack of vertical $C_T$* observations, we tested two alternative approximations of ID, which are:

- Mixed layer depth (MLD)

- Temperature penetration depth (TPD)

MLD and TPD are described in detail in Sect. 2.6.3. The two parameterisations were further applied to following two test data sets, both of which contain the required surface $pCO_2$ and vertically resolved temperature and salinity data:

- In situ data from the BloomSail campaign without $pCO_2$ data at depth (SV Tina V (surface only))

- Combined SOOP surface $pCO_2$ observations and modelled salinity and temperature profiles (SOOP Finnmaid + GETM model)

For both data sets, $C_T{}^*$ time series were calculated based on the same observed mean $A_T$ as used to derive the NCP best-guess (Sect. 2.4). Please note that neither the MLD nor the TPD approach allows to resolve vertical entrainment fluxes, because
profiles of $C_T{}^*$ are not reconstructed (compare section 2.5.3). Based on all possible combinations of two $C_T{}^*$ time series and two integration depth constraints, four reconstructed NCP time series were derived and compared to the best–guess (i.e. the estimate based on the vertically resolved $pCO_2$ observations from this study).

### 2.6.1 SOOP *Finnmaid* surface $pCO_2$

SOOP *Finnmaid* regularly commutes between Helsinki in Finnland and Travemünde in Germany thereby crossing the entire
Central Baltic Sea and our study area on the east coast of Gotland every $1 - 2$ days. On board SOOP *Finnmaid*, $pCO_2$ is measured with a bubble–type equilibrator system supplied with water from an inlet at around 3 m water depth. Details of the measurement set–up are described in Schneider et al. (2014) and data are submitted on a regular basis to the Surface Ocean $CO_2$ Atlas SOCAT (Bakker et al., 2016). The primary measurement system used to determine $pCO_2$ in this study is a NDIR sensor (LI-6262, LI-COR Biosciences, Lincoln, USA). The ferrybox unit is also equipped with an additional methane/carbon dioxide
analyzer (Greenhouse Gas Analyzer DLT 100, type 908-0011, Los Gatos Research, San Jose, USA), providing independent $pCO_2$ observations (Gülzow et al., 2011). Intercomparison of both systems is routinely used to ensure the correct functioning of the instrumentation. In this study, a data gap caused by malfunctioning of the primary LI-COR system was filled by including data recorded with the Los Gatos system on six cruises between July 8 and 16 (see Appendix D for details). The mean regional $pCO_2$, sea surface temperature (SST) and salinity (SSS) were calculated for each crossing of the study area (Fig. 1b). Based
on the mean $pCO_2$ and SST values, $C_T{}^*$ was calculated following the procedure outlined in Sect. 2.4. A remaining gap in the SOOP time series was filled with two in situ $C_T{}^*$ observations from the BloomSail campaign (July 19 and 24).

### 2.6.2 GETM model temperature and salinity

Surface SOOP measurements were complemented with vertically–resolved salinity and temperature data from the output of a numerical ocean model of the Baltic Sea. The deployed General Estuarine Turbulence Model (GETM) has a horizontal reso-
285 lution of 1 nautical mile and 50 vertical terrain–following levels. The uppermost level has a thickness of maximum 50 cm to

properly represent SST and ocean–atmosphere fluxes. The computation of the atmospheric fluxes is based on the parameterisation of Kara et al. (2005). The model covers the entire Baltic Sea and the period 1961 – 2019. A detailed analysis of the model performance is given in Placke et al. (2018) and Gräwe et al. (2019). For the present study, we used a model run restarted in 2003 with the atmospheric forcing from the operational reanalysis data set of the German weather service (Zängl et al., 2015).

Additionally, we implemented the Langmuir–circulation parameterisation of Axell (2002), to account for wind–wave induced variation in the mixed layer depth. Model results were averaged over 24 h and interpolated to a standardised section with 2 km horizontal and 1 m vertical resolution, which follows the mean *Finnmaid* cruise track. Based on this standard section, daily mean profiles within the study area (characterized by little regional variability) were computed and linearly interpolated to match the exact times of *Finnmaid* crossings.

### 2.6.3 Parameterisation of the integration depth (ID)

In this study, two parameters were used to integrate surface observations across depth, namely the classical mixed layer depth (MLD) and the newly introduced temperature penetration depth (TPD).

MLD was defined as the shallowest depth at which seawater density exceeds the density at the surface by more than $0.1$ kg m$^{-3}$ (Roquet et al., 2015). According to this definition, MLD characterises the thermohaline structure of the water column and often (but not necessarily) approximates the depth to which surfaces water masses are actively mixed. The definition through a fixed density threshold further implies that gradual changes of temperature with depth are not reflected by this parameter.

TPD characterises the mean penetration depth of surface warming that occurred between two sampling events. TPD was defined as the integrated warming signal across the water column, i.e. the sum of all positive temperature changes within 1m depth intervals, divided by the SST increase (for a graphical illustration see Fig. C4a). According to this definition and in contrast to MLD, TPD takes gradual changes of temperature across depth into account and does not require a fixed threshold value. TPD is only applicable when SST increases and has units of metres. To illustrate the TPD concept, it should be noted that a homogeneous warming signal that ceases abruptly at 10 m water depth would result in the same TPD as a warming signal that decreases linearly from the surface to 20 m water depth (TPD is 10 m in both cases). The TPD approach is motivated by the assumption that primary production and temperature increase are both primarily controlled by the light dose that a water parcel received (Schneider et al., 2014) and therefore show similar vertical patterns.

In analogy to TPD, the penetration depth of $C_T$* drawdown (CPD) was defined as the integrated loss of $C_T$* across the water column divided by the decrease of $C_T$* at the surface (Fig. C4b).

# 3 Results

## 3.1 Dynamics of temperature, $p\text{CO}_2$, $C_T$* and phytoplankton biomass

Between July 6 and August 16, a total number of 78 complete vertical CTD and $p\text{CO}_2$ downcast profiles were recorded (Fig. 2 and 3). $C_T$* was calculated and profiles were regionally averaged for each of the eight cruise events (Fig. 4). Since the first cruise of the BloomSail expedition on July 6, sea surface temperature (SST) increased steadily from ~15 °C to peak values of ~25 °C (Fig. 4 and 5) observed on August 3. Sea surface $p\text{CO}_2$ was already as low as ~100 $\mu$atm at the beginning of July (Fig. 5a) and decreased further to the lowest values of ~70 $\mu$atm on July 24. The drop in $p\text{CO}_2$ and the simultaneous increase in SST correspond to a decrease of $C_T$* of almost 90 $\mu$mol kg$^{-1}$ (Fig. 4). During this period of intense primary production, the regional variability of SST, $p\text{CO}_2$, and $C_T$* across stations was low compared to their temporal change (Fig. 5a–b; Fig. C3). The regional variability is slightly higher when including the coastal stations 01, 13, and 14 (results not shown), but is generally lower than one could expect from the bloom patchiness typically observed through remote sensing (Fig. 1a). With respect to $p\text{CO}_2$ dynamics, it should be noted that (i) the observed temperature increase and $C_T$* drawdown have opposing effects on $p\text{CO}_2$ and (ii) the change of $p\text{CO}_2$ per change in $C_T$* is generally low at low absolute $p\text{CO}_2$. The observed $C_T$* dynamics in surface waters are clearly attributable to the primary production activity of phytoplankton and go along with an observed increase of the biomass of *Nodularia* sp. (Fig. B2), which also peaked on July 24. Furthermore, we found that $C_T$* calculated from $p\text{CO}_2$ agreed with $C_T$* derived from discrete samples within the uncertainty range attributed to regional variability (Fig. 5c).

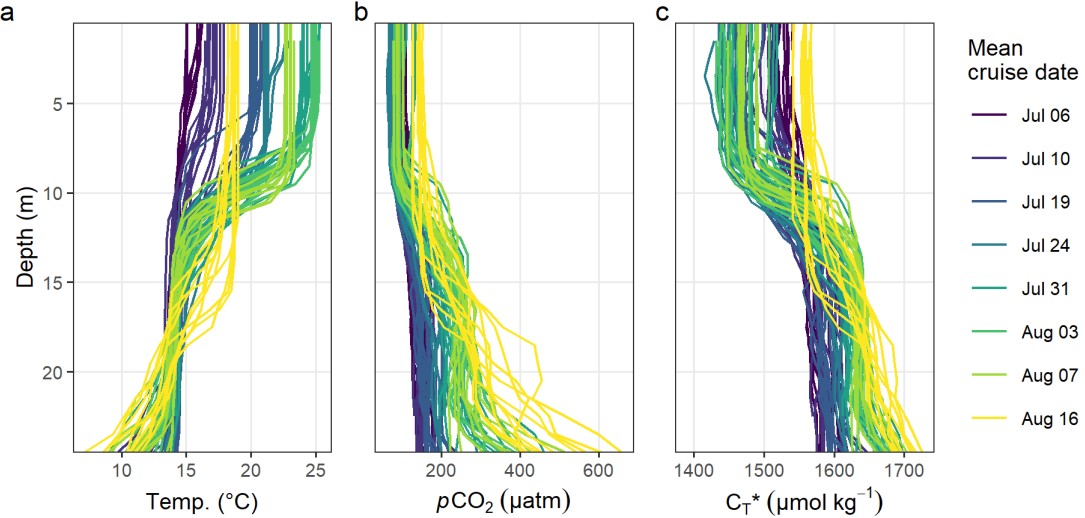

**Figure 3.** Overview of (a) temperature, (b) $p\text{CO}_2$ and (c) $C_T$* profiles recorded throughout the BloomSail field sampling campaign on board SV *Tina V*.

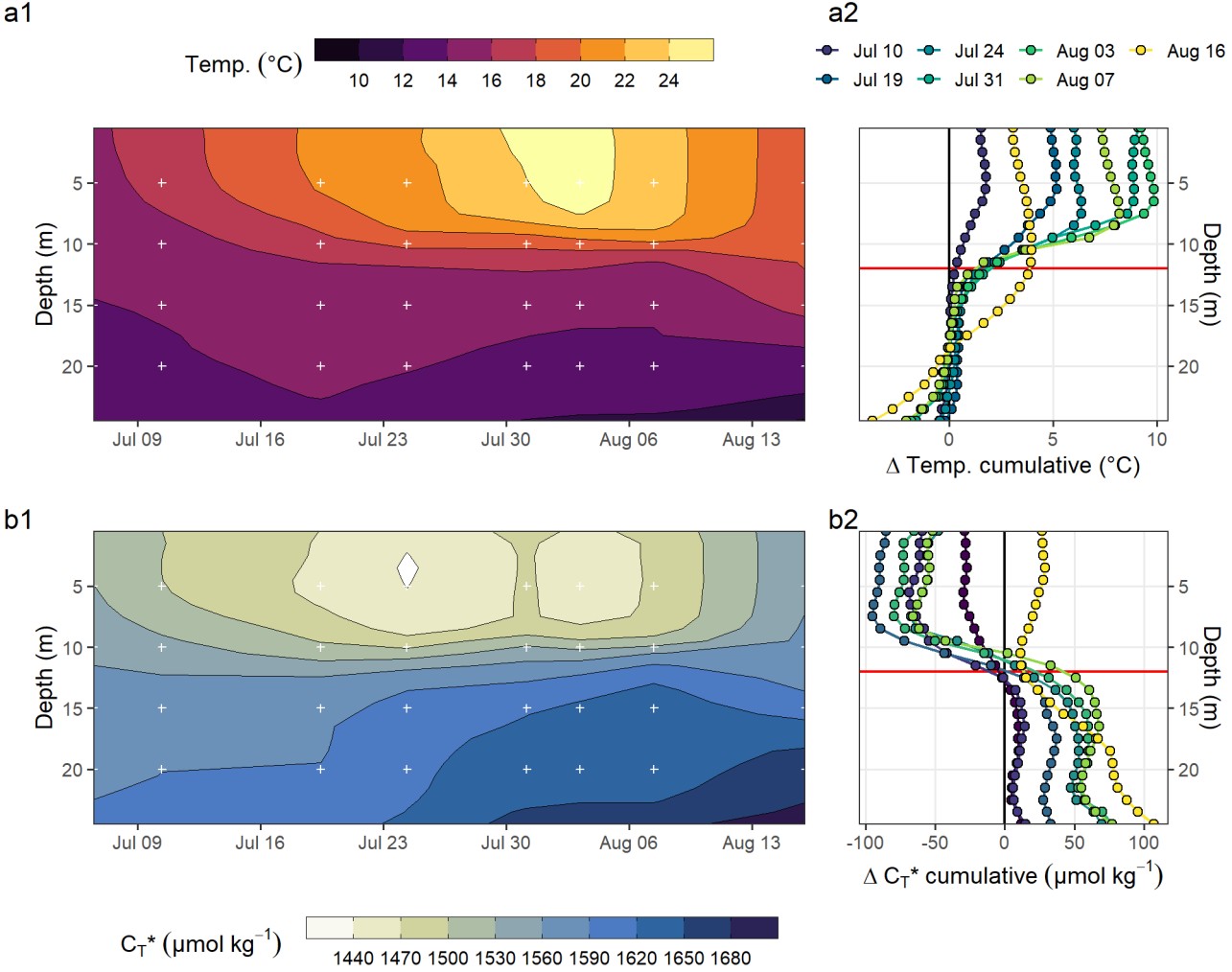

**Figure 4.** (a) Temperature and (b) $C_T^*$ between July 6 and August 16 displayed as (1) Hovmoeller plots and (2) profiles of cumulative changes since the first cruise on July 6. Mean cruise dates are indicated by the horizontal position of white symbols in (a1) and (b1), and the compensation depth of 12 m is indicated as a red, horizontal line in (a2) and (b2).

Between the extremes of $p\mathrm{CO_2}$ and $C_T^*$ (minimum on July 24) and SST (maximum on August 3), a noticeable increase of surface $C_T^*$ was observed on July 31, which was accompanied by a higher regional variability across the station network (Fig. 5a,c). The temporary $C_T^*$ increase was limited to the north–eastern stations 07 – 10 (Fig. C3) and paralleled by a drop in salinity and elevated $A_T$ at the same stations (Fig. B1). It is therefore attributable to the lateral exchange of water masses. All signals of this lateral intrusion vanished within a week. At the other stations (02 – 06 and 11 – 12), no noticeable signs of water mass exchange or $C_T^*$ changes were observed between July 24 and August 3, indicating that the production and respiration of

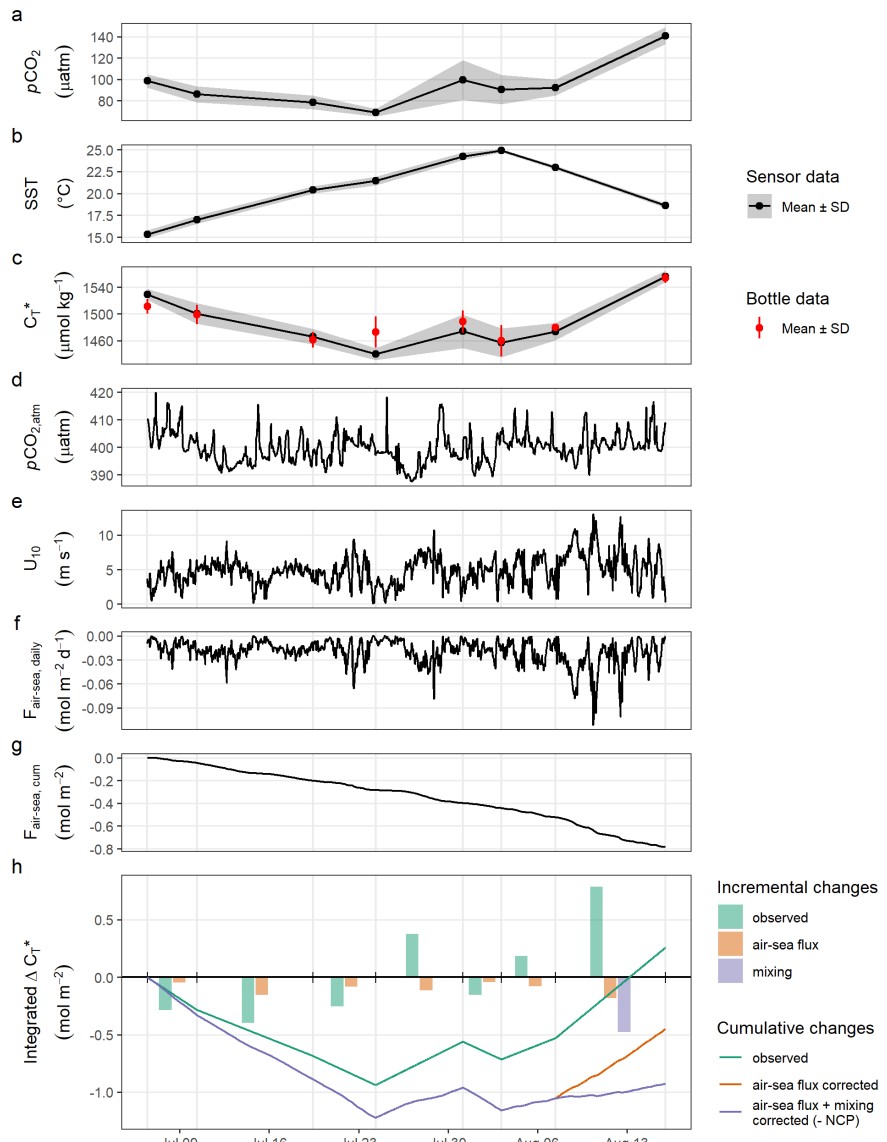

**Figure 5.** Time series displaying from top to bottom: Sea surface observation of (a) $p\mathrm{CO_2}$, (b) temperature, (c) $\mathrm{C_T}^*$ with grey ribbons indicating the standard deviation across stations and red symbols representing discrete sample data; atmospheric observations of (d) $p\mathrm{CO_{2,atm}}$, (e) wind speed at 10 m, (f) daily and (g) cumulative air–sea fluxes of $\mathrm{CO_2}$; as well as (h) the derived water column inventory changes of $\mathrm{C_T}^*$. In (h), bars represent incremental changes between cruise events (vertical grid lines), whereas lines represent cumulative changes since the first cruise. Colours distinguish observed $\mathrm{C_T}^*$ changes from values referring to the applied air–sea $\mathrm{CO_2}$ flux and mixing correction. Net community production (NCP) is equal to the negative value of flux and mixing corrected cumulative changes of $\mathrm{C_T}^*$ (purple line) and peaks on July 24.

organic matter were balanced during this period. During the first two weeks of August the study area was affected by increased wind speeds, causing a decrease of SST back to ~18 °C. The simultaneous return of surface $pCO_2$ to ~150 $\mu$atm corresponded to a $C_T$* increase of ~100 $\mu$mol kg$^{-1}$.

The observed surface warming and $C_T$* drawdown extended vertically to a water depth of ~10 m (Fig. 4). On the first cruise day (July 6), the vertical distribution of $C_T$* and temperature was still relatively homogeneous. $C_T$* at 25 m water depth was ~70 $\mu$mol kg$^{-1}$ higher than at the surface. Likewise, the temperature gradient covered only ~3 °C from 16 °C at the very surface to 13 °C at depth. The warming of surface waters caused an increasingly stable thermocline to be established at around 10 m water depth, which reached a temperature gradient of ~10 °C across 5 m on August 3. Likewise, continuous and uniform

drawdown of $C_T$* within the surface layer enhanced the vertical $C_T$* gradient to >150 $\mu$mol kg$^{-1}$ between the surface and 25 m water depth. The $C_T$* drawdown was observed to a maximum depth of 12 m.

    Between August 7 and 16 the SST drop of ~6°C was accompanied by a temperature increase in deeper water layers (11 – 17 m) of up to 5 °C. This vertical redistribution of heat indicates vertical mixing of water masses, which was also reflected in a steep increase of $C_T$* in the surface water and a loss of $C_T$* between 11 – 17 m (Fig. 3 and 4).

**3.2   NCP best–guess based on profiling measurements**

Net community production (NCP) was determined through vertical integration of the observed drawdown of $C_T$* from the surface to the compensation depth located at 12 m. The determined compensation depth reflects the maximum penetration depth of the incremental (i.e. between cruise days), as well as the cumulative (i.e. from July 6 – 24), $C_T$* drawdown (Fig. 4). Likewise, about 95% of the cumulative warming signal, which refers to positive temperature changes integrated over depth,

occurred above 12 m.

    Until July 24, the depth–integrated $C_T$* drawdown amounted to ~0.9 mol m$^{-2}$ (Fig. 5h). This observed $C_T$* drawdown was corrected for air–sea fluxes of $CO_2$. Between July 6 and August 7, the cumulative air–sea flux ($F_{air\text{-}sea, cum}$) amounted to around -0.5 mol m$^{-2}$ (Fig. 5g), with a negative sign representing $CO_2$ uptake from the atmosphere. In the absence of noticeable vertical mixing, this flux was entirely added to the observed $C_T$* drawdown. Only between August 7 and 16, when mixing to

360 about 17 m water depth was observed, a significant fraction of the $CO_2$ taken up from the atmosphere was transported below 12 m water depth. To account for the partial loss of airborne $CO_2$ to deeper waters during this 9 day–period, only 12/17 of $F_{air\text{-}sea, cum}$ during this time (-0.2 mol m$^{-2}$), which is the fraction that would remain in the upper water column, was added to the observed $C_T$* drawdown. In addition, a significant amount of $C_T$* entrainment (-0.5 mol m$^{-2}$) into the surface layer was caused by the vertical mixing between August 7 and 16 (Fig. 5h and C2).

After correction for air–sea fluxes and vertical entrainment of $CO_2$, the cumulative changes of depth–integrated $C_T$* represent the NCP between the sea surface and the compensation depth at 12 m (Fig. 5h). The peak NCP value of ~1.2 mol m$^{-2}$ was observed on July 24 and is of primary interest because it reflects the amount of organic matter that was produced and is potentially available to be either exported or remineralised. The temporary drop in the NCP best-guess on July 31 is due to the lateral exchange of water masses as described in Sect. 3.1. Deriving the NCP time series without the stations affected by

later exchange of water masses (07–10) results in an almost identical NCP estimate on July 24, but a reduced drop on July 31

(data not shown). In both cases, no signs of continued NCP were observed after July 24. Accordingly, our interpretation of the reconstructed NCP based on surface $pCO_2$ observations will focus on the comparison to the peak value of our NCP best–guess on July 24.

### 3.3 NCP reconstruction based on surface $pCO_2$ and hydrographical profiles

The reconstruction of depth–integrated NCP was tested for two data sets containing the same type of information, namely the observed changes in surface $pCO_2$ and vertical profiles of seawater salinity and temperature. The first data set "SV Tina V (surface only)" contains the surface $pCO_2$ data recorded during the BloomSail expedition, as well as the complete CTD profiles. The second data set ("SOOP Finnmaid + GETM model") combines surface $pCO_2$ observations from SOOP *Finnmaid* with seawater salinity and temperature as estimated with the GETM model.

An almost identical decrease of surface $C_T$* of ~50 $\mu$mol kg$^{-1}$ was determined between July 6 and 16 (Fig. 6a), based on the completely independent $pCO_2$ data recorded on SOOP *Finnmaid* and SV *Tina V*. Likewise, a very similar increase in $C_T$* between August 6 and 15 was determined from both independent observational data sets. The good agreement between the independent observations justifies that a data gap due to failure of instrumentation on the SOOP was filled with two observations from SV *Tina V* on July 19 and 24 (open circles in (Fig. 6a).

Good agreement was also found for the spatio–temporal dynamics of observed and modelled seawater temperature (Fig. 6b). Observed and modelled SST agreed within 1 °C over the entire observation period, despite an absolute change spanning almost 10 °C. Slightly higher deviations between observed and modelled temperature were found around the thermocline, where the observational record revealed a stronger temperature gradient. This difference is likely due to an imperfect representation of Langmuir circulation in the model (Axell, 2002), whereas the absence of increased light attenuation caused by phytoplank-390 ton particles was previously found to have only minor impacts on modeled SST dynamics (Löptien and Meier, 2011). Most importantly, the mean temperature penetration depths (TPD) derived from the observational and model data differ less than 1 m, indicating that surface warming and the integrated heat uptake are accurately represented by the model. The TPD (mean $\pm$ SD) over the observed productive period between July 6 and 24 was determined as 12.3 $\pm$ 2.5 m and 11.4 $\pm$ 2.3 m for the observational and model data, respectively (Fig. 6b). The TPD estimates are considerably higher than the respective mixed 395 layer depth (MLD) estimates (6.0 $\pm$ 1.9 m and 5.5 $\pm$ 1.2 m) and agree better with the observed penetration depth of $C_T$* drawdown, indicating that TPD is the favourable parameterisation of the integration depth.

Based on SOOP observations before July 6, first signs of the onset of the investigated bloom event were detected already on July 3. Between July 3 and 6, an SST increase of ~1 °C was accompanied by a $C_T$* drawdown of ~10 $\mu$mol kg$^{-1}$ (data not shown). Still, in the absence of any vertically resolved observation for this time period, the following comparison of the 400 reconstructions to the best–guess needs to be restricted to the period July 6 – 24, during which the bulk of NCP occurred.

The NCP reconstruction based on TPD is generally higher than the MLD–based estimate (Fig. 6c). Comparing peak cumulative NCP estimates for July 24, the TPD–approach results in a ~10% overestimation compared to the best–guess, i.e. the value derived from vertically resolved measurements. In contrast, the MLD–based NCP estimate is ~30% lower than the best–guess.

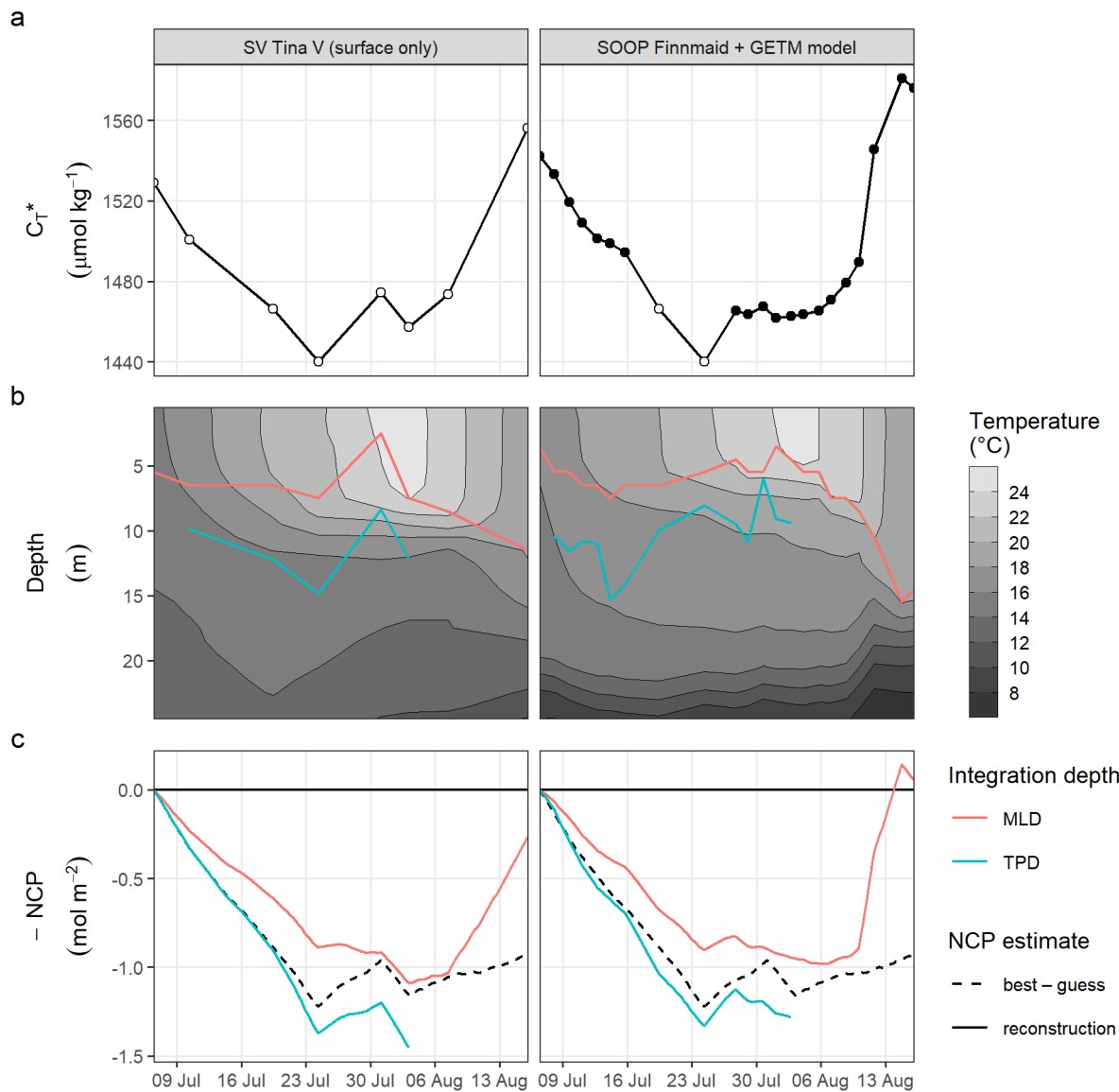

**Figure 6.** Time series illustrating the reconstruction of depth–integrated NCP from surface $C_T*$ and vertically resolved hydrographical parameters. Displayed are results based on two test data sets, namely observations from SV *Tina V* without $C_T*$ data at depth (left panels) and a combination of SOOP $C_T*$ and modelled hydrographical data (right panels). From top to bottom, panels represent (a) surface $C_T*$, (b) the vertical distribution of temperature together with the mixed layer depth (MLD) and temperature penetration depth (TPD) for each cruise day, and (c) depth–integrated NCP comparing the reconstructions (solid lines) with the best–guess (dashed black line) according to Fig. 5. Please note that a data gap in the SOOP record was filled with two observations from SV *Tina V* (open circles in a).

The reconstructed NCP estimates are very similar for both test data sets, as the good agreement between the underlying $C_T^*$, MLD and TPD time series suggests.

Comparing the deviation between the best–guess and reconstructed NCP estimates in the light of the lateral variability observed within the study area, it must be emphasised that between July 6 and 24, the mean standard deviation of $pCO_2$ and $C_T^*$ across stations amounted to $\pm$ 6 $\mu$atm and $\pm$ 11 $\mu$mol kg$^{-1}$, respectively. This is higher than the likely uncertainty associated with the $pCO_2$ measurements (see Methods), as well as its response time correction (see Methods and Appendix A4) or conversion to $C_T^*$ (see Appendix C1). Therefore, the lateral variability of seawater chemistry and the production signal are generally considered the highest source of uncertainty to our NCP estimates. Still, this lateral variability is small compared to the signal to be resolved (i.e. the $C_T^*$ drawdown of ~90 $\mu$mol kg$^{-1}$). However, on a relative scale the lateral $C_T^*$ variability is about as large as the difference between the best–guess and the TPD–based NCP reconstruction (~10%), suggesting that the bias of the reconstruction falls within the uncertainty range of the best-guess. In contrast, the lateral variability is smaller than the deviation between the best–guess and the MLD–based NCP reconstruction.

All reconstructed NCP estimates include the correction of air–sea fluxes of $CO_2$, but it is impossible to quantify and correct vertical entrainment fluxes due to mixing, because the vertical distribution of $C_T^*$ across the water column can not be resolved. The strong deviation between the best–guess NCP and the MLD-based reconstruction on August 16 is due to this missing correction of vertical mixing. This deviation highlights that the reconstruction approach is only applicable to production periods with a stable or shoaling thermocline. The TPD-based approach does not allow for any estimate during the last two weeks of the observations period, as the TPD is per definition only applicable to periods of warming surface waters.

## 4 Discussion

### 4.1 Comparison to previous studies

Having in mind the application of our NCP reconstruction approach to surface $pCO_2$ observation collected since 2003, it is important to examine if the biogeochemical dynamics of the examined cyanobacteria bloom in 2018 is representative for those in other years. Unfortunately, only a few previous studies aimed at the quantification of cyanobacteria growth as a component of the Baltic Sea carbon budget. One exception is the interpretation of SOOP *Finnmaid* data by Schneider et al. (2014). Focusing on the period from June to August and taking into consideration individual production pulses observed in the years 2005, 2008, 2009 and 2011, the authors found average daily rates of $C_T^*$ drawdown ranging from 3 to 8 $\mu$mol kg$^{-1}$ d$^{-1}$, which comprises the mean rate of 5 $\mu$mol kg$^{-1}$ d$^{-1}$ determined in this study (i.e. the average $C_T^*$ drawdown of ~90 $\mu$mol kg$^{-1}$ over 18 days, Fig. 4). The individual production events identified by Schneider et al. (2014) lasted 1 to 5 weeks, similar to the bloom duration described in this study. Finally, Schneider et al. (2014) also provided a depth–integrated NCP estimate based on daily modelled mixing depths, which ranged from 3 – 20 m and were derived from the vertical distribution of a tracer one day after its injection into the surface. Although this approach is primarily useful to estimate the vertical distribution of air–sea $CO_2$ fluxes and does not necessarily reflect the vertical extent of organic matter production, their determined mid–summer NCP estimates (1 – 2.1 mol m$^{-2}$) are in the same order of magnitude as the best–guess derived in this study. It should be noted that the NCP estimates

by Schneider et al. (2014) refer to the cumulative NCP of one to three production pulses per years, whereas our estimate of ~1.2 mol m$^{-2}$ refers to a single bloom event.

Wasmund et al. (2001) conducted $^{14}$C incubation experiments at different water depths to determine instantaneous rates of daytime primary production during a cyanobacteria bloom. Their reported carbon fixation rates in surface waters (0.4 – 0.8 mmol C m$^{-3}$ h$^{-1}$) are in the same order of magnitude as the mean rate found in this study (5 $\mu$mol kg$^{-1}$ d$^{-1}$, equivalent to 0.2 mmol C m$^{-3}$ h$^{-1}$), despite representing daytime production rates and diurnal averages, respectively. More important than the agreement between the fixation rates at the sea surface, is the fact that Wasmund et al. (2001) also found significantly lower fixation rates below 10 m water depth (< 0.2 mmol–C m$^{-3}$ h$^{-1}$), which agrees well with the depth distribution of NCP observed in this study.

Furthermore, the succession of different cyanobacteria genera observed in 2018, with the *Nodularia* dominated bloom following an earlier presence of *Aphanizomenon* (Fig. B2), was previously described as a typical pattern (Wasmund, 2017), as well as the fact that increased wind speed and turbulence can inhibit N–fixation of cyanobacteria and cause the termination of the bloom (Wasmund, 1997).

In conclusion, the bloom event duration, $C_T$* drawdown, and NCP, as well as the vertical extend of carbon fixation and the succession of the bloom observed in this study agree well with observations in previous years, and distinct differences cannot be found. We therefore conclude that the findings of this study are representative for Baltic Sea cyanobacteria blooms in general, although the SST and $p$CO$_2$ levels in 2018 were at the upper and lower end, respectively, of the conditions observed in previous years (Schneider and Müller, 2018).

## 4.2 Biogeochemical relevance and interpretation

Our best–guess of cumulative NCP on July 24 (~1.2 mol m$^{-2}$) represents the net amount of organic matter that was produced throughout the bloom event in the surface waters above the compensation depth at 12 m. After subtracting ~20 % dissolved organic carbon (DOC) production (Hansell and Carlson, 1998; Schneider and Kuss, 2004), our NCP estimate equals the produced particulate organic carbon (POC) that is potentially available for export. In contrast, NCP estimates derived from other traditional methods for the integration across depth (such as the lower bound of the euphotic zone or the mixed layer depth) would not directly relate to the POC export potential.

However, the potential POC export constraint by our NCP estimate is not equivalent to the supply of organic matter to the deep waters of the Gotland Basin, because POC might be (partly) remineralised before sinking beneath the permanent halocline. Remineralisation of POC that occurs during the bloom event above the compensation depth is – according to our definition of NCP – already included in our estimate. In contrast, any additional remineralisation of POC that occurs between the compensation depth and the halocline, or above the compensation depth after the end of the bloom event, reduces the organic matter supply to the deep waters and thereby mitigates deoxygenation. Indeed, our profiling measurements indicate a steady accumulation of $C_T$* beneath the compensation depth (Fig. 4), likely fueled by the remineralisation of organic matter. However, our measurements do neither allow to constrain the budget of this $C_T$* accumulation, nor could we attribute the source of organic matter.

In contrast to shallow remineralisation processes, the deepening of the mixed layer that marked the end of the studied bloom event may facilitate the efficient transport of POC from the surface layer to depth. Focusing on the accumulation of remineralisation products beneath 150 m in the Gotland basin, a previous study revealed that – in accordance with the main input of POC during the productive period – remineralisation rates exhibit a pronounced seasonality (Schneider et al., 2010). This seasonality was found to be most pronounced in the water layers closest to the sediment surface, suggesting that beneath 150 m the remineralisation takes place mainly at the sediment surface and is of minor importance during particle sinking through the deep water column. The pronounced seasonality further confirms that surface organic matter production and deep water oxygen consumption are indeed tightly coupled, despite a potential degradation of POC before export across the permanent halocline.

We conclude that NCP estimates determined with the methods developed in this study are of direct relevance to quantify the drivers for deep water deoxygenation. However, a better understanding of the organic matter remineralisation processes would be required to close the budget of biogeochemical transformations. New observational platforms, such as recently deployed biogeochemical ARGO floats (Haavisto et al., 2018), will complement the existing SOOP infrastructure and help to provide the required observational constraints throughout the water column.

### 4.3 Recommendations and caveats for NCP reconstruction from SOOP and model data

The good agreement between our best–guess and the TPD-based NCP reconstruction on July 24 (Fig. 6c) indicates that it is possible to determine NCP from surface $p$CO$_2$ observations and vertically resolved seawater temperature with little uncertainty. For the NCP calculation based on surface $p$CO$_2$ observations from SOOP and modelled temperature profiles, we recommend to:

1. Convert surface $p$CO$_2$ to C$_T$* based on a mean A$_T$ estimate for the region under consideration.

2. Identify production pulses dominated by cyanobacteria as periods characterised by a decrease in C$_T$* that occurs between June and August.

3. Integrate observed surface C$_T$* changes to the temperature penetration depth (TPD) estimated from modelled temperature profiles, rather than using a mixed layer depth (MLD) estimate.

4. Perform the integration individually for each production pulse and limit NCP reconstruction to periods characterised by a stable or shoaling thermocline.

It should be emphasised that lateral variability and water mass transport are critical for observation–based NCP estimates and constitute the largest source of uncertainty in our estimates. However, SOOP observations allow averaging of observations across large regions, which reduces the impact of lateral water mass transport (Schneider and Müller, 2018). The region for spatial averaging should be chosen large enough to avoid as much as possible the influence of lateral perturbations which depend on the surface dynamics and the biogeochemical gradients in the surrounding area. Yet, the region for spatial averaging should be chosen small enough to ensure that variations of $p$CO$_2$ within the region are small compared to the temporal changes

of interest. Another critical aspect of the recommended NCP reconstruction approach is the restriction to periods of a stable or shoaling thermocline. While in principle it is possible that net organic matter production could occur also during periods of
505 a deepening thermocline, this process was neither observed in this study nor in previous years (Schneider and Müller, 2018), and is in line with findings from the long–term cyanobacteria monitoring program unraveling that increased wind speed causes the termination of the bloom (Wasmund, 1997). We thus conclude that reconstructed NCP estimates are not affected by a systematic underestimation due to this temporal restriction. Likewise, the required mean $A_T$ estimate should not restrict the applicability of our approach even if $A_T$ is not directly measured. For the Baltic Sea, it was demonstrated (Schneider et al.,
2003) that $A_T$ estimated from the known $A_T$–S relationship (Müller et al., 2016) is sufficiently accurate to convert $p\mathrm{CO}_2$ to $C_T$* (see also Appendix C1).

The NCP reconstruction approach presented in this study was derived from observations covering a single bloom event within the Central Baltic Sea. In the lack of comparable comprehensive observational data that underlie our best–guess, the applicability of this approach could not be tested for other regions or bloom events. However, the dynamics and intensity of
515 the bloom event described here are comparable to previous, independent descriptions of cyanobacteria blooms. Therefore, it is assumed that underlying biogeochemical mechanisms are representative and that the NCP reconstruction approach can be applied to other cyanobacteria bloom events. Specifically, we assume that the findings represented here can be applied to evaluate past and future $p\mathrm{CO}_2$ observations made on *Finnmaid* and other SOOP in the Central Baltic Sea without compromise. However, larger uncertainties should be expected when applying our NCP reconstruction approach to other regions outside the
520 Central Baltic Sea.

## 5 Conclusions

In this study, the depth–integrated quantification of NCP that occurred during a cyanobacteria bloom in the Baltic Sea in 2018 is achieved through the interpretation of profiling measurements of $p\mathrm{CO}_2$ that covered the entire bloom event. Furthermore, it is demonstrated that this best–guess can be reconstructed with small bias from SOOP $p\mathrm{CO}_2$ observations and modelled
temperature profiles. Recommendations to apply our reconstruction approach to the comprehensive long–term record of surface $p\mathrm{CO}_2$ data available for the Baltic Sea are given. The application of this approach will allow for the detection and attribution of trends in cyanobacteria NCP over decades. In particular the comparison of NCP estimates of bloom events that occurred under different environmental conditions will provide a better understanding of the controlling factors. Factors to be tested include the environmental parameters used to constrain NCP ($p\mathrm{CO}_2$, SST, and TPD), but also additional observations of nutrients
and phytoplankton composition routinely determined on SOOP Finnmaid and in the framework of the Baltic Sea monitoring program. The recently started initiative to deploy biogeochemical ARGO floats in the Baltic Sea will further aid to link surface NCP estimates and deep water deoxygenation, and thereby constrain biogeochemical budgets in the Baltic Sea. Ultimately, this knowledge will inform the design and monitoring of conservation measures aiming at a Good Environmental Status of the Baltic Sea and potentially other regions.

*Code and data availability.*

**Website**: Following the concept of literature programming and relying on the R package workflowr (Blischak et al., 2019), the code, plain text comments, and graphical output of this study are compiled as a website available at: https://jens-daniel-mueller.github.io/BloomSail/.

**Code and raw data**: A release of the Github repository underlying the website and containing all code was tagged as "bg-2021-40_resubmission" and archived on https://zenodo.org/. All raw data required to run the analysis were uploaded manually to this archive. Thus, the combined code and data are available under doi: https://doi.org/10.5281/zenodo.4553314.

**Processed environmental data**: Processed in situ observation of this study will be made available through https://www.pangaea.de/ upon acceptance of the manuscript.

## Appendix A: $pCO_2$ sensor measurements

### A1    Sensor calibration

The CONTROS HydroC® $CO_2$ sensor used in this study (serial number $CO_2$-0618-001) was calibrated in water by the manu-
facturer at 15 °C before (June 2018) and after (October 2018) the deployment for a measuring range of 100 to 500 $\mu$atm. The
pre– and post–deployment calibration polynomials met the 6 steps per calibration with an $R^2$ of 0.999999 (pre) and 0.999993
(post) at an RMSE of 0.13 $\mu$atm (pre) and 0.43 $\mu$atm (post). The time between the calibrations was about 107 days and the
sensor runtime during this interval was about 506 hours or little more than 21 days. The zero drift observed between the two
calibrations was only 0.89 $\mu$atm.

### A2    Sensor configuration and operation

The instrument periodically records zeroing values, during which the $CO_2$ within the gas stream is scrubbed by a soda lime
cartridge. Zeroings of two minutes duration were recorded every five hours during the field deployment. A period of 600
seconds after the zeroing was flagged as a flush period, during which the sensor signal recovers to environmental conditions.
Recordings during the flush and zeroing period were removed before further biogeochemical interpretation.

For the majority of the measurements, the sensor was operated with a 8W–pump (SBE-5T; Sea-Bird Electronics, Bellevue,
USA) and the logging interval was set to 1 second. Only for the first two cruise days on July 6 and 10, a 1W–pump (SBE-5M,
Sea-Bird Electronics) was used and the logging interval set to 10 seconds.

The downcast profiles were always recorded continuously and with a steady profiling speed of ~2 m min$^{-1}$. The upcast
profiles were either performed continuously as well, or with a stop to record an equilibrated reference $pCO_2$ value at a desired
depth. Only continuous downcast profiles were used for biogeochemical interpretation.

Zeroing signals were recorded by the CTD unit from the analogue sensor output, as well as in the internal sensor memory.
Both records were used to ensure exact temporal match of the CTD and $pCO_2$ time series. Only $pCO_2$ data stored with higher
temporal resolution in the internal memory were used during further analysis.

### 565    A3    Data post–processing

A drift correction as discussed in Fietzek et al. (2014) was applied to the field data to improve the data quality. This post–
processing considers information from the pre– and post–deployment calibrations (i.e. concentration dependent or span drift)
and the regular in situ zeroings (i.e. zero drift).

The first 60 seconds within every zeroing interval were discarded to only consider smooth zero–gas measurements that are
not affected by the signal drop from ambient $pCO_2$ to the zero value. Zero signals for every point of the deployment were
obtained by linear interpolation of the zero measurements. In case of data gaps larger than 2 hours within the deployment
data, the course of the 2 zero signals before or after the gap was linearly extrapolated forward or backward, respectively,
instead of an interpolation over the time of the measuring gap. A concentration–dependent drift of the sensor was considered

by transforming the pre– into the post–deployment calibration polynomial according to the actual sensor runtime (and not according to the course of the zero measurements as applied within Fietzek et al. (2014)).

Approx. 100 unrealistic outliers were found within the sensor temperature record ($T_{sensor}$ parameter) of the HydroC®. These were identified to be electronic artefacts and the values replaced by the constant temperatures recorded before and after these events that only lasted a few seconds at most.

## A4 $p\text{CO}_2$ response time correction

The $p\text{CO}_2$ response time correction applies a common "growth-law equation" and follows a two-step procedure (Miloshevich et al., 2004) that was previously successfully applied to $p\text{CO}_2$ data recorded with the same type of instrument as used in this study (Fiedler et al., 2013; Fietzek et al., 2014; Atamanchuk et al., 2015). In a first step, the actual in situ response times ($\tau$) of the sensor were determined by fitting an exponential function to the signal recovery following a zeroing (Sect. A4.1). In a second step, the determined $\tau$ values were used to correct the signal delay (Sect. A4.2). A quality assessment of the $p\text{CO}_2$ response time correction is given in Sect. A4.3.

### A4.1 Response time determination

In situ response times ($\tau$) were determined from $p\text{CO}_2$ data recorded during the flush period after each zeroing. Data recorded during the initial 20 seconds of each flush period were removed as those are affected by the mixing of residual gas volumes inside the sensor. Individual $\tau$ values were determined by fitting the non–linear model

$$p\text{CO}_2(t) = p\text{CO}_2(t_{end}) + (p\text{CO}_2(t_0) - p\text{CO}_2(t_{end})) \cdot e^{(-dt/\tau)} \tag{A1}$$

where $p\text{CO}_2(t)$ is the recorded $p\text{CO}_2$ at time t, $p\text{CO}_2(t_0)$ and $p\text{CO}_2(t_{end})$ are the fitted $p\text{CO}_2$ values at the beginning and the end of the equilibration process, and $dt$ is the time since the beginning of the equilibration process. In situ $\tau$ was determined for a fit interval length of 300 seconds. Flush periods were discarded when the mean of absolute residuals from the fit exceeded 1% of the final $p\text{CO}_2$, a condition which indicated unstable environmental $p\text{CO}_2$ (e.g. due to unintended heaving of the sensor package).

Similar to previous studies, a decrease of $\tau$ with increasing in situ temperature was found. The dependence of $\tau$ on temperature was fitted with linear regression models, separately for the deployments with the 1W– and 8W–pump. The sensor was carefully cleaned after each cruise and no signs of a changing sensor response time over time as an indicative of fouling on the sensor's membrane were detected.

### A4.2 Correction procedure

For each recorded $p\text{CO}_2$ value, the corresponding $\tau$ was calculated from measured in situ temperature. The response time correction was then applied based on a rearranged version of equation A1:

$$p\text{CO}_{2,insitu}(t_{i+1}) = \frac{p\text{CO}_{2,obs}(t_{i+1}) - p\text{CO}_{2,obs}(t_i) \cdot e^{(-dt/\tau)}}{1 - e^{(-dt/\tau)}} \tag{A2}$$

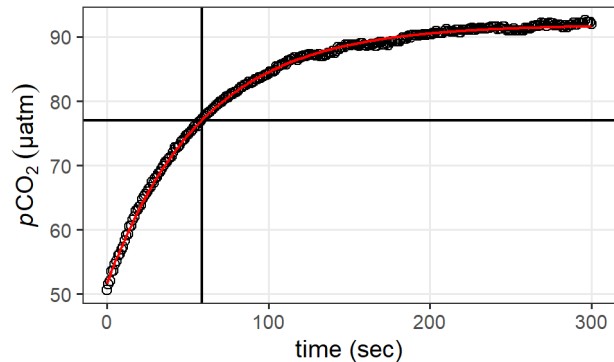

**Figure A1.** Exemplary determination of the response time $\tau$ through fitting an exponential function (red curve) to the $pCO_2$ signal recovery following a zeroing measurement. The determined response time $\tau$ and $pCO_2(t=\tau)$ are indicated by a vertical and horizontal line, respectively.

where $pCO_{2,insitu}$ is the true in situ $pCO_2$ time series, $pCO_{2,obs}$ the $pCO_2$ time series as recorded by the sensor, and $\tau$ the response time for the interval between $t_i$ and $t_{i+1}$. A rolling mean with a window width of 30 sec was applied to the response time corrected $pCO_{2,insitu}$ time series to remove short term noise. Please note that throughout the rest of the manuscript $pCO_{2,insitu}$ is referred to as $pCO_2$.

### A4.3 Quality assessment

The improvements by the response time correction were investigated based on the difference between up- and downcast $pCO_2$ profiles vertically gridded into 1m depth intervals. To focus this quality assessment on the conditions in near surface waters which are subject of this study, profiles were discarded which exceeded a maximum depth of 30 m and/or a maximum $pCO_2$ of 300 $\mu$atm. Those profiles were excluded only for the quality assessment (not for the biogeochemical interpretation) to avoid a bias through exposure to very high $pCO_2$ at greater depth. Furthermore, profiles were removed with a maximum number of missing observations from two or more depth intervals, which occasionally occurred when a sensor zeroing started while profiling. Based on this subset of response time corrected $pCO_2$ profiles it was found that the mean absolute $pCO_2$ difference between the up– and downcast profile was <2.5 $\mu$atm averaged across the upper 5 m of the water column and <7.5 $\mu$atm across the upper 20 m. The highest offset was found at around 10 m water depth and results from the steep environmental $pCO_2$ gradient around the thermocline.

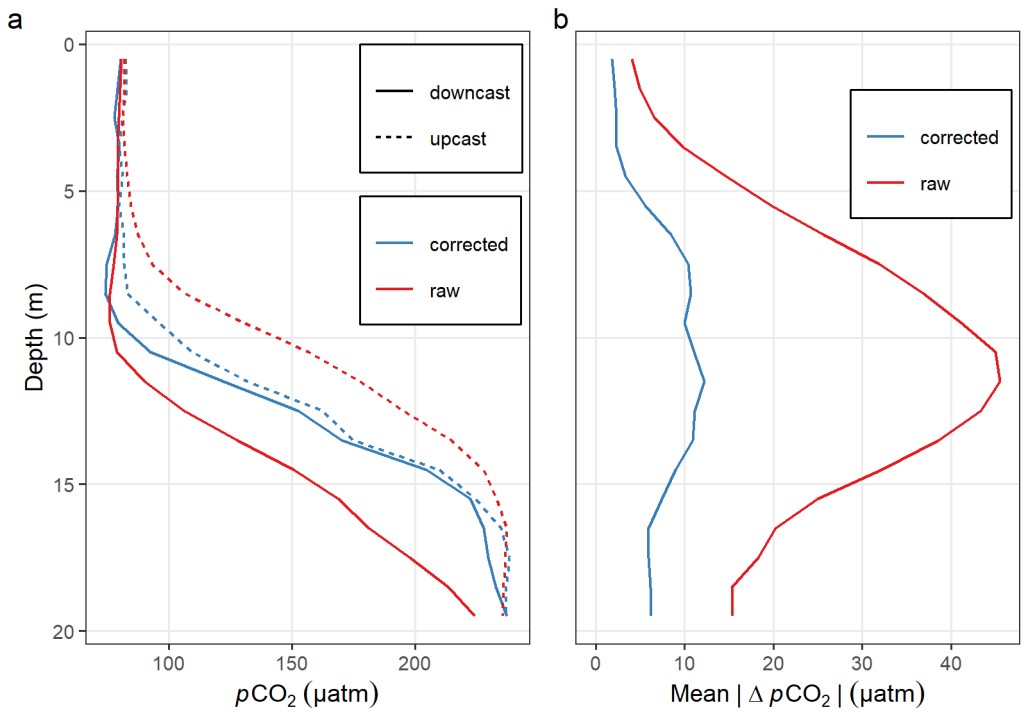

**Figure A2.** Comparison of $p\mathrm{CO_2}$ profiles before (raw) and after (corrected) response time correction: (a) Exemplary up– and downcast $p\mathrm{CO_2}$ profiles at one station and (b) mean absolute $p\mathrm{CO_2}$ difference between up– and downcast profiles across all profiles included in the quality assessment.

## B1   $C_T$ and $A_T$

Dissolved inorganic carbon ($C_T$) was determined from discrete bottle samples with an Automated Infra Red Inorganic Carbon Analyzer (AIRICA, MARIANDA, Kiel, Germany). The analysis relies on the stripping of $CO_2$ through acidification. The released $CO_2$ is transported with a nitrogen carrier gas stream to an infrared detection unit (LI-7000, LI-COR Biosciences, Lincoln, USA), where the peak area is determined. Comparison to measurements performed on certified reference materials (CRM Batch 173; Dickson et al., 2003) allows for the calculation of $C_T$. Triplicated measurements were performed on each sample and a precision of 2 $\mu$mol kg$^{-1}$ was achieved.

Total alkalinity ($A_T$) was analysed by open cell titration of $125 - 140$ g of sample. The method involves a two–stage titration. After a first, single addition of hydrochloric acid to achieve a pH $4 - 3.5$, $A_T$ is determined during a continued, stepwise titration to pH 3, during which pH is recorded potentiometrically (Dickson et al., 2007). Measurements were referenced to CRM batch 173 (Dickson et al., 2003).

$C_T^*$ calculated for discrete samples refers to a classical alkalinity–normalised $C_T$, and was defined as $C_T^* = C_T \cdot A_{T,mean}$ / $A_T$. $C_T^*$ derived from discrete samples or $p$CO$_2$ sensor data are directly comparable (Fig. 5c) because they are referenced to the same mean $A_T$ of the discrete samples (1720 $\mu$mol kg$^{-1}$).

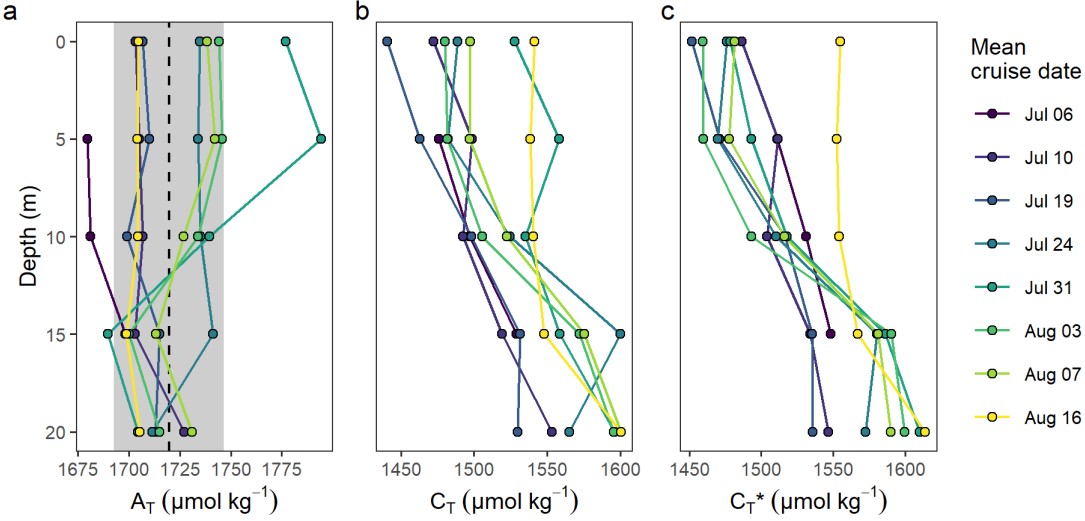

**Figure B1.** Vertical profiles of (a) $A_T$, (b) $C_T$, and (c) $C_T$ normalised to the mean alkalinity ($C_T^*$). Shown are cruise mean values for discrete samples taken at stations 07 and 10. The dashed line and grey area in (a) indicate the mean $\pm$ 1 standard deviation of $A_T$ across the upper 20 m of the water column.

## B2 Phytoplankton

Phytoplankton samples were fixed with Lugol solution within no more than 24 hours after sampling. Samples were stored dark, before being transported to IOW and analysed in the laboratory within no more than 3 months after sampling. Phytoplankton community composition and biomass were determined by the Utermöhl method (HELCOM, 2017), which relies on microscope counts and the conversion of cell shape and size to biomass units.

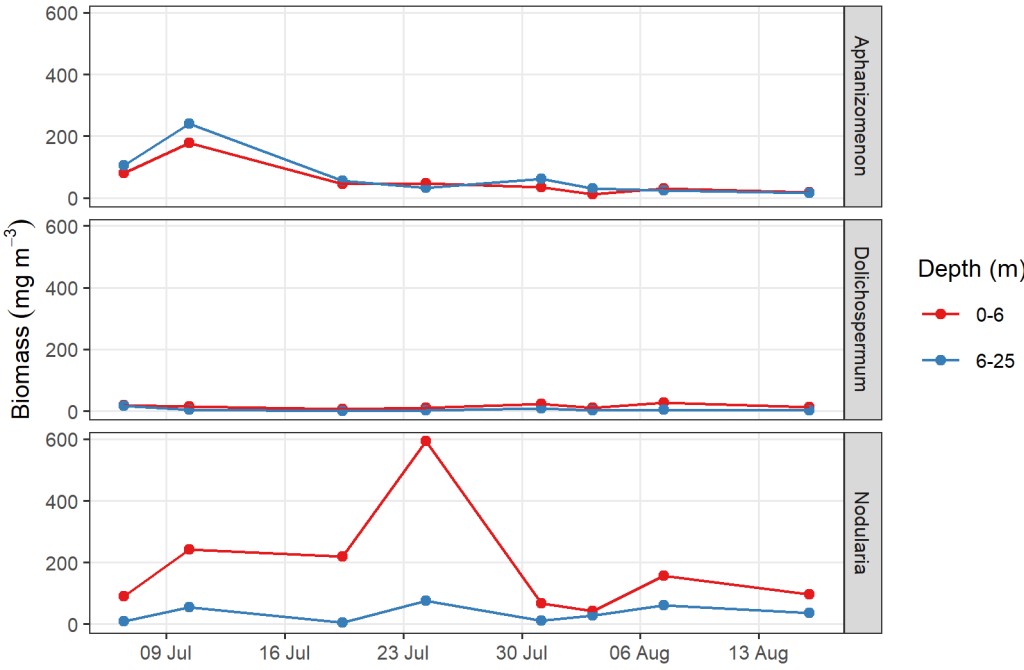

**Figure B2.** Time series of cyanobacterial biomass, averaged for surface (0 – 6 m) and subsurface (6 – 25 m) water masses sampled from stations 07 and 10 (Fig. 1). Results are based on microscope counts and distinguish three genera (panels).

## Appendix C: Net community production estimation

### C1 Conversion from $pCO_2$ to $C_T$*

The approach to estimate temporal changes (rather than absolute values) in the dissolved inorganic carbon concentration ($C_T$) from a $pCO_2$ time series was previously established and theoretically examined (Schneider et al., 2014, and references therein). It relies on a fixed estimate of alkalinity ($A_T$) and is only applicable when noticeable internal changes in $A_T$ can be excluded, as is the case in the Baltic Sea due to the absence of calcifying plankton (Tyrrell et al., 2008). To avoid confusion with measured or absolute $C_T$ values and for consistency with previous studies, the calculated variable is referred to as $C_T$*.

To evaluate the applicability of this approach under the specific $pCO_2$ and temperature conditions observed in summer 2018, we calculated $C_T$* changes between Jul 6 and 24 for a range of $A_T$ values covering three times the standard deviation of $A_T$ observations (Fig. B1). For assumed $A_T$ values of 1747 $\mu$mol kg$^{-1}$ and 1693 $\mu$mol kg$^{-1}$, which is 1 standard deviation of the observations (27 $\mu$mol kg$^{-1}$) higher and lower than the mean $A_T$ (1720 $\mu$mol kg$^{-1}$), the bias of the derived change in $C_T$* amounts to $\pm$ 1.6 $\mu$mol kg$^{-1}$. This bias is <2% compared to the signal of interest, i.e. the absolute drawdown of $C_T$* (89 $\mu$mol kg$^{-1}$).

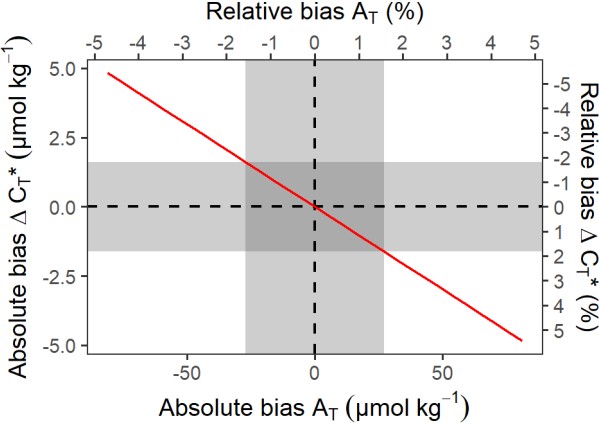

**Figure C1.** Bias of changes in $C_T$* as a function of the bias in mean $A_T$ used for calculation (see Fig. B1). Results correspond to the $pCO_2$ and temperature conditions observed in this study and are expressed in absolute and relative units. Grey areas highlight $\pm$1 standard deviation around the mean $A_T$.

It should be noted that the bias assessment presented here reflects two types of errors, namely (i) the assignment of an erroneous mean $A_T$ value for the calculation and (ii) the lateral exchange of water masses with different $A_T$ but identical initial $pCO_2$ during the observation period. The robustness of this approach to the latter aspect is the reason why $pCO_2$ observations are more suitable to determine NCP than direct $C_T$ measurements, when those are not normalised to corresponding $A_T$ measurements.

## C2 Calculation of the vertical entrainment flux of $C_T*$

The vertical entrainment flux of $C_T*$ that occured across the 12 m integration depth layer between Aug 7 and 16 was estimated assuming an instantaneous complete vertical mixing to 17 m water depth after Aug 7. For this scenario, the hypothetical homogeneous $C_T*$ concentration after the mixing event ($C_T*mix$) equals the mean volume–weighted $C_T*$ concentration between 0 – 17 m (Fig. C2). Furthermore, the entrainment flux ($C_T*flux$) into the surface water column (0 – 12 m) is equal to the concentration difference between observed $C_T*$ on Aug 7 and $C_T*mix$, integrated from 12 to 17 m.

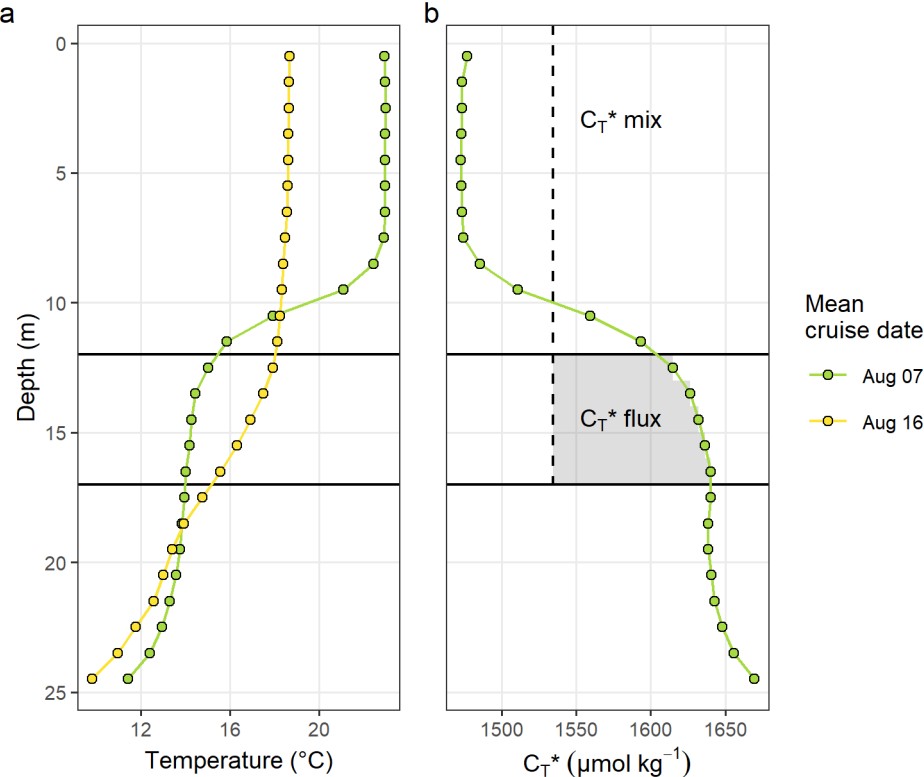

**Figure C2.** Illustration of the approximation of the entrainment flux of $C_T*$ due to vertical mixing. (a) The estimated deepening of the mixed layer from 12 to 17 m water depth between Aug 7 and 16 is based on the observed changes in the temperature profiles. (b) Assuming a complete, instantaneous mixing of the water column after Aug 7, the hypothetical homogeneous concentration of $C_T*$ ($C_T*mix$) can be used to approximate the entrainment flux of $C_T*$ (grey area).

## C3    Individual stations profiles

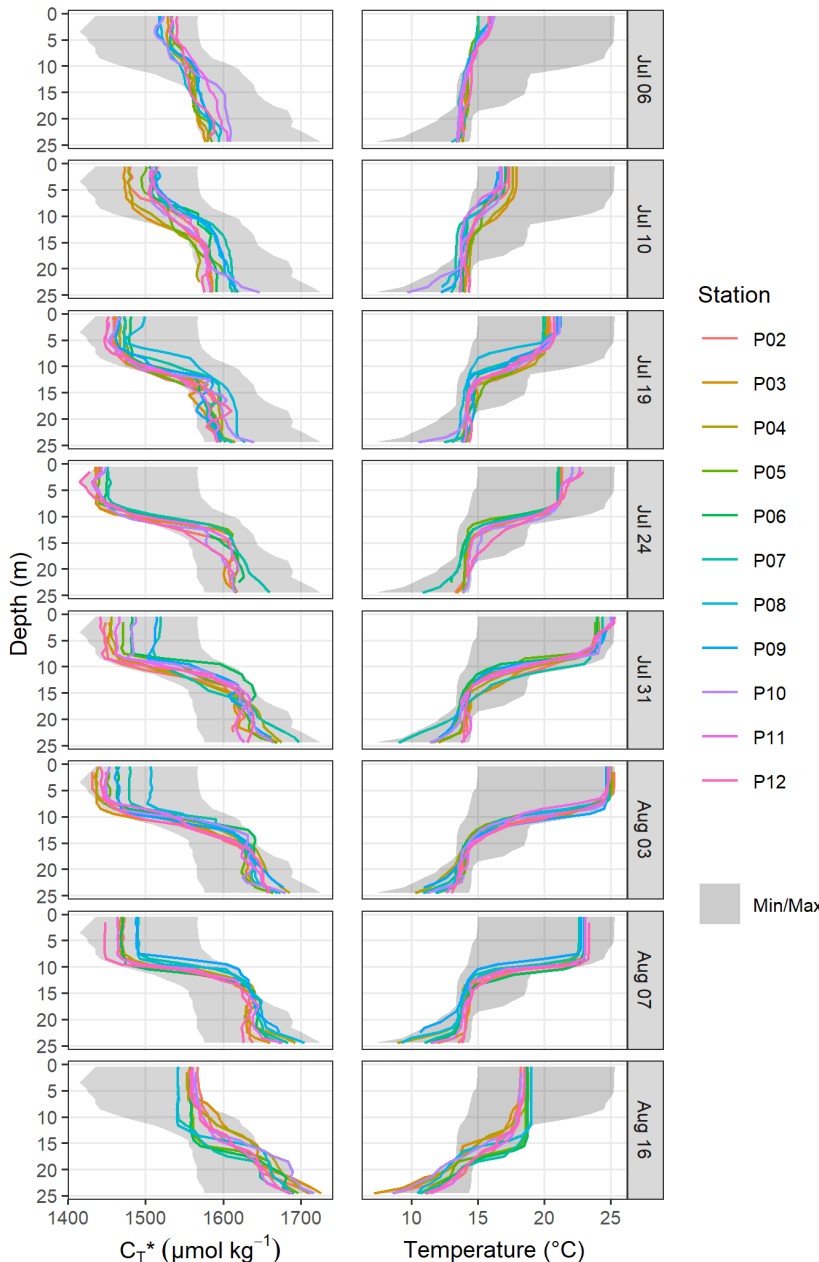

**Figure C3.** Individual profiles of $C_T$* (left panels) and temperature (right) displayed separately for each cruise day (rows) and station (color). Grey ribbons indicate the minimum and maximum values observed over the entire study period.

## C4 Temperature penetration depth (TPD) concept

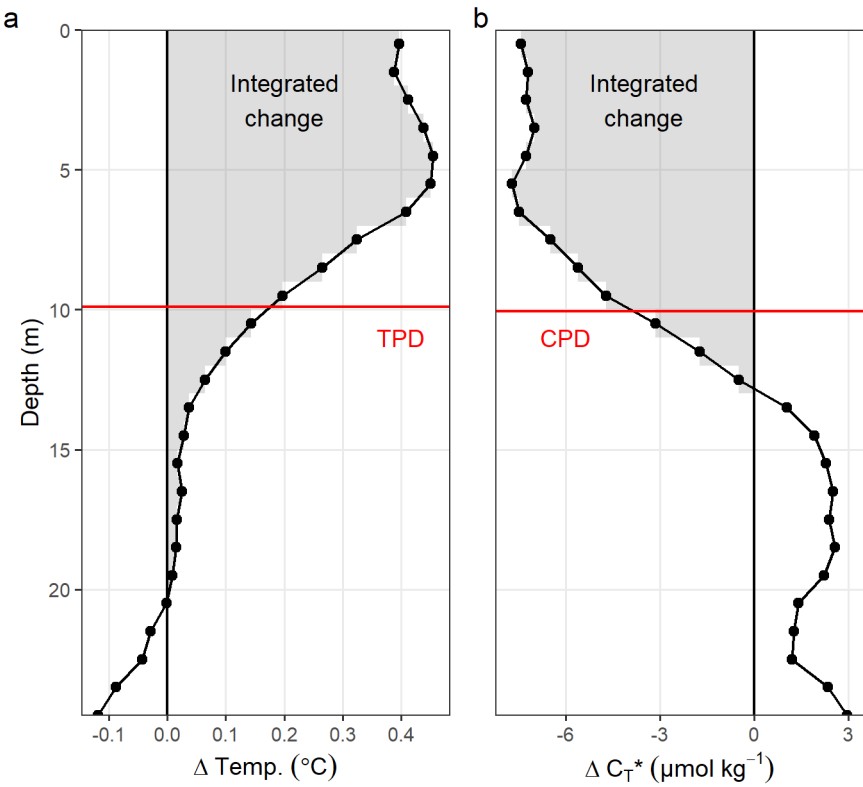

**Figure C4.** Illustration of the temperature and $C_T*$ penetration depth concept, short TPD and CPD. Shown are exemplary profiles of incremental changes of (a) temperature and (b) $C_T*$ observed between the cruises on July 6 and 10. TPD and CPD (red horizontal lines) are defined as the depth–integrated positive (for temperature) and negative (for $C_T*$) changes (grey areas) divided by the change at the surface. TPD and CPD are expressed in units of metres.

## Appendix D: SOOP *Finnmaid* $p\text{CO}_2$

For SOOP *Finnmaid* transects recorded between July 7 and July 16, $p\text{CO}_2$ data were not available from the LI-COR system because of technical failure. Therefore, data generated by the Los Gatos (LGR) system were used to fill the gap. Unfortunately, the comparison of LI-COR and LGR measurements before July 7 indicated a small leakage in the LGR system, which was later also physically detected and fixed. The resulting difference between the two systems was clearly correlated with absolute $p\text{CO}_2$, as expected from contamination with ambient air. For data from the transect on July 5, the linear regression model $p\text{CO}_{2,true}$ = $p\text{CO}_{2,LGR}$ + 0.038 * $p\text{CO}_{2,LGR}$ - 24.2 was fitted, assuming that the LI-COR system had delivered the "true" $p\text{CO}_{2,true}$ before its failure. Assuming further that the effect of the contamination remained constant, this relationship was then applied to reconstruct $p\text{CO}_{2,true}$ from $p\text{CO}_{2,LGR}$ for the period without LI-COR data. To validate this adjustment, $p\text{CO}_{2,true}$ was also reconstructed from $p\text{CO}_{2,LGR}$ on July 4 and compared to $p\text{CO}_2$ directly measured with the LI-COR system. The mean difference was below 2 $\mu$atm for the entire transect as well as for a data subset within the study region, giving confidence to the high accuracy of the adjusted $p\text{CO}_{2,true}$. It should be noted that the adjusted SOOP $p\text{CO}_2$ data recorded between July 7 and July 16 agree well with the in situ $p\text{CO}_2$ recorded by the sailing campaign, i.e. the standard deviations of all surface measurements in the study region overlap.

*Author contributions.* JDM was the lead author of this study and involved in all parts of it, in particular the conceptualization, funding acquisition, project administration, field sampling, data curation, visualisation and analysis, and manuscript writing. BS was involved in the conceptualization, formal analysis, and original draft preparation. UG generated and processed the GETM model data. PF supported the HydroC® sensor configuration and performed the data post processing. MBW and AR provided the atmospheric data and supported the field campaign logistically. NW was responsibly for the collection, analysis and interpretation of phytoplankton samples. SK designed and assembled the sensor package. GR was involved in the conceptualization, funding acquisition, and project administration and supervision. All authors contributed to the internal review and editing of the manuscript.

*Competing interests.* The authors declare that they have no conflict of interest.

*Acknowledgements.* The BloomSail expedition would not have been possible without the tremendous support of a fantastic group of people. In particular, 13 sailors supported the activities at sea. In the order of presence on board, those are Friederike Saathoff, Yannic Wocken, Andreas Kubatzki, Amanda Nylund, Rainer Kroiß, Ben Ole Grabler, Anna von Brandis, Fynn Pasewald, Ronja Topp, Matthias Kreuzburg, Katharina Klehmet, Mareike Körner, and Bastian Wulf. Monika Gerth supported the expedition by providing regular remote sensing products. Stefan Otto, Jenny Jeschek and Paul Cwierz supported the analysis of discrete samples. Furthermore, the authors are thankful for logistic and administrative support received from IOW (Beatrix Blabusch, Barbara Hentzsch, Kristin Beck) as well as in the harbour of Herrvik (Mats Hanson, Kenneth Broscheit). Technical equipment of the sailing vessel was supported among others by Segelwerkstatt Warnemünde, Lopolight, Delius Klasing, BMS, and Kadematic. We acknowledge the use of imagery from the NASA Worldview application (https://worldview.earthdata.nasa.gov/), part of the NASA Earth Observing System Data and Information System (EOSDIS). Furthermore, JDM acknowledges the funding of an early career grant received from the National Geographic Society through grant number EC-178R-18, as well as inhouse funding received from IOW. The research leading to this manuscript has received funding from BONUS, the joint Baltic Sea research and development programme (Art 185), funded jointly from the European Union's Seventh Programme for research, technological development and demonstration and from the German Federal Ministry of Education and Research through Grant No. Grant No. 03F0773A (BONUS INTEGRAL). Measurements on SOOP Finnmaid are a German contribution to the Integrated Carbon Observation System (ICOS), and received funding through the German Federal Ministry of Education and Research and the German Federal Ministry of Transport and Digital Infrastructure. The ICOS station Östergarnsholm is funded by the Swedish Research Council and Uppsala University. We acknowledge detailed and very constructive suggestions from two anonymous reviewers.

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
