# Peer review of "Cyanobacteria net community production in the Baltic Sea as inferred from profiling $pCO_2$ measurements"

_Biogeosciences, 2021_

## Referee Comment (RC1)

Review of '**Cyanobacteria net community production in the Baltic Sea as inferred from profiling $p$CO₂ measurements**' by Mueller et al.

The manuscript describes the application of the water budget method to constrain inorganic carbon drawdown, a proxy for Net Community Production (NCP), during the cyanobacteria bloom in the Baltic Sea. The work also examines the possibility of making use of the long-term ferry-box measurements of surface pCO2 for the hindcasting of NCPs. The authors argue that the quantification of NCP is key to understanding the biogeochemical controls of cyanobacteria blooms in the Baltic Sea. The authors present compelling arguments substantiated by robust methodological approaches and error examinations. Overall, this is a well-written and well-structured work with scientific merit and is worthy of publication in *Biogeosciences.* Said that there are certain areas where the manuscript can be improved, and I would suggest minor revisions aiming at strengthening the message of this manuscript.

General major comments (no particular order):

- Traditionally NCP is constrained to the depth of the euphotic zone, compensation depth or MLD. Differences between these and their implications for the export of organic matter should be explained and discussed, even though you chose to use TPD in your approach.
- Discuss how NCP is related to export production and deoxygenation. Which processes and phenomena do alter/modulate organic matter supply to the deep?
- It is not clear how getting NCP for the previous years will aid in better understanding the bloom controls since there have been no simultaneous (to SOOP operation) studies of bloom phenology, POC profiles or nutrient concentrations.
- Which measurements, on which platforms and on which scales will help you 'disentangling the drivers of NCP' in the future.
- Please discuss the early-spring bloom and its contribution to the annual NCP.
- In the paper you mention that your approach to NCP (using surface pCO2 and the modelled MLDs/TPDs) is limited to the time-periods of a stable or shoaling thermocline. Would you go as far as to estimate the % of time when the bloom falls outside this pattern based on your modelled MLD/TPD? This could give you some information/confidence about the applicability of the NCP approach using surface pCO2 only in the lack of full profiles.
- The authors may want to include the calculations of heat flux and use it as a proxy for CT* drawdown. These will likely correlate.

Specific comments:

L15: specify units of NCP (mol m^-2 (?))

L20: within 10% (if taking into account the period of a stable thermocline?) please specify

L30: see the general comments re conventional definition of NCP. Please add the missing info here.

L33: 1.2.Baltic Sea: please add more references citing studies of drivers and causes of hypoxia/anoxia in the Baltic Sea and sealed estuaries in general.

L47: how important is the spring bloom in the Baltic Sea relative to the summer bloom? Which species dominate during the spring bloom?

L58: 'The limited understanding of the factors…' This study does not explicitly address the factors controlling the bloom, instead examines the ways to quantify NCP. This paragraph needs rewording to include background information relevant to *this* study.

L73: cite the literature examining equilibration time of gases.

L75-L80: do these statements refer to the Baltic Sea? Need to make this clear to the reader. Global N-based NCP estimates are very popular)

L136: what are the accuracies and precision of the sensors after their calibrations?

L148: 'of 1% of reading'. Is this true for this particular study? Did you compare $pCO_2$ (sensor) to $pCO_2$ (DIC-TA)? If so, this comparison should be included instead.

L149: Please justify using A4.1 for tau determination and A4.2 for the correction? Why not to stick to one equation? Both work great, although A4.2 add more noise to the data.

L159: based on (the length) of the release line??

L160: $HgCl_2$

2.4.1 should go before L 176

L177: in my opinion, the term 'estimate' already refers to an indirect, non-empirical way to assign value to a certain phenomenon. 'Best-guess' should, therefore, be dropped.

L276: 'Since July $6^{th}$….' what portion or % of the production you expect prior to July $6^{th}$?

L293: 'that NCP has ceased'…do you mean there is no production (NPP), nutrient limitation or that NPP is balanced by respiration?

Fig. 5 f)-g) Fair-sea or Fatm for clariyy on Y-axis

h) flux = air-sea flux. 'Flux' is too ambiguous in this case.

flux mixing corrected ( - NCP ) – production has a positive sign, so negative to integrated CT*

Fig.6. c) again, NCP must be positive by conventional definition. Either use NCP(CT) or some other way to clarify why Net Community Production in your case is negative.

L400: Please also explain how to overcome the caveats. This info can be useful for the reader. What other measurements are needed?

L420: Indeed, it is unlikely that production will continue during the periods of deepening thermocline. However, deepening-restratification events may have an impact on both entrainment of CT and detrainment of POC. The latter, especially, can moderate more efficient transport from the photic zone to the depth bypassing the respiration stage. Therefore, it appears crucial to monitor the frequency of occurrence of such events as they impact both the production and efficiency of carbon export.

---

## Author Response (AR1)

**AC1: Reply on RC1**

Provided by Jens Daniel Müller on behalf of all co-authors

**Dear Referee 1,**

**Thank you for providing your review, which we considered very helpful to strengthen the presentation of our study. Most of your comments request an extension of the manuscript with additional information, which we are happy to implement. In particular, we aim to add a dedicated discussion section addressing the biogeochemical interpretation of our NCP estimates. Several points you raised were also in agreement with RC2, and we cross reference our answers where this applies.**

**Please find our detailed answers (bold font) and proposed text edits (bold italic font) next to your comments (normal font) below. Line numbers refer to the initially submitted version of the manuscript.**

**We hope to have addressed all of your comments appropriately, but welcome additional feedback if required.**

**Best wishes**
**Jens Daniel Müller, on behalf of all co-authors**

General major comments (no particular order):

Traditionally NCP is constrained to the depth of the euphotic zone, compensation depth or MLD. Differences between these and their implications for the export of organic matter should be explained and discussed, even though you chose to use TPD in your approach.

**We will introduce traditional NCP concepts by inserting following text in L30 of the introduction:**
***"... we define NCP as the net amount of carbon fixed in organic matter (gross production minus respiration) that is produced in a defined water volume over a defined period. This definition implies that the choice of an integration depth is a critical component of any NCP estimate. Traditionally, NCP is constrained either to the depth of the euphotic zone, the compensation depth at which gross production equals respiration, or the mixed layer depth (Sarmiento and Gruber, 2006). Of those approaches, only the integration to the compensation depth is directly linked to the vertical distribution of carbon fixation and remineralisation and therefore quantifies the amount of formed organic matter that can potentially be exported."***

**To align the terminology used to describe our methods with the description of traditional concepts, we will clarify in L180 that our best-guess is constrained to the compensation depth:**

*"... we derive the column inventory of incremental changes of $\Delta C_T^*$ ($i\Delta C_T^*$) between two cruise events through vertical integration of $\Delta C_T^*$ from the sea surface to the compensation depth (cd), i.e. the depth (z) at which no net drawdown of $CO_2$ was observed"*

**This clarification in the Methods will further be supported by an equation summarizing our NCP calculation, as proposed by R2.**

**We will further clarify in L214 that only for our NCP reconstructions, we replace the traditional compensation depth with a MLD/TPD constraint:**
*"In the lack of vertical $C_T^*$ observations that would allow us to determine the compensation depth, we tested two alternative approximations of the integration depth, which are: ..."*

**Finally, we will add a dedicated section in L399 of the discussion to describe the biogeochemical relevance and interpretation of the chosen NCP constraint with respect to organic matter export:**
*"Our best–guess of cumulative NCP on July 24 (~1.2 mol m$^{-2}$) represents the net amount of organic matter that was produced throughout the bloom event in the surface waters above the compensation depth at 12 m. After subtracting ~20 % dissolved organic carbon (DOC) production, our NCP estimate equals the produced particulate organic carbon (POC) that is potentially available for export. In contrast, NCP estimates derived from other traditional methods for the integration across depth (such as the lower bound of the euphotic zone or the mixed layer depth) would not directly relate to the POC export potential."*
**This additional information was also requested by RC2.**

Discuss how NCP is related to export production and deoxygenation. Which processes and phenomena do alter/modulate organic matter supply to the deep?

**In the new section that will be included in L399 to discuss the biogeochemical relevance and interpretation of our findings, we will address this question as follows:**
*"However, the potential POC export constraint by our NCP estimate is not equivalent to the supply of organic matter to the deep waters of the Gotland Basin, because POC might be (partly) remineralised before sinking beneath the permanent halocline. Remineralisation of POC that occurs during the bloom event above the compensation depth is – according to our definition of NCP – already included in our estimate. In contrast, any additional remineralisation of POC that occurs between the compensation depth and the halocline, or above the compensation depth after the end of the bloom event, reduces the organic matter supply to the deep waters and thereby mitigates deoxygenation. Indeed, our profiling measurements indicate a steady accumulation of $C_T^*$ beneath the compensation depth (Fig. 4), likely fueled by the remineralisation of organic matter. However, our measurements do neither allow to constrain the budget of this $C_T^*$ accumulation, nor could we attribute the source of organic matter."*
**This additional information was also requested by RC2.**

It is not clear how getting NCP for the previous years will aid in better understanding the bloom controls since there have been no simultaneous (to SOOP operation) studies of bloom phenology, POC profiles or nutrient concentrations.

**We will clarify this by adding the following information to our conclusions (L437):**
*"The application of this approach will allow for the detection and attribution of trends in cyanobacteria NCP across decades. In particular the comparison of NCP estimates of bloom events that occurred under different environmental conditions will provide a better understanding of the controlling factors. Factors to be tested include the environmental parameters used to constrain NCP ($pCO_2$, SST, and TPD), but also additional observations of nutrients and phytoplankton composition routinely determined on SOOP Finnmaid and in the framework of the Baltic Sea monitoring program. The recently started initiative to deploy biogeochemical ARGO floats in the Baltic Sea will further aid to link surface NCP estimates and deep water deoxygenation, and thereby constrain biogeochemical budgets in the Baltic Sea."*
**This additional information was also requested by RC2.**

Which measurements, on which platforms and on which scales will help you 'disentangling the drivers of NCP' in the future.

**Please refer to the reply above.**

Please discuss the early-spring bloom and its contribution to the annual NCP.

**Enabling reliable NCP estimates for the mid-summer cyanobacteria bloom is the core aim of this study. A reliable hindcast of the mid-summer NCP will only be possible when the findings of this study are applied to almost two decades of available SOOP $pCO_2$ data. In the absence of this information, the assessment of the importance of the spring relative to the mid-summer bloom is highly uncertain. Nevertheless, we agree that some more information on the spring-bloom and a rough approximation of its contribution to the annual NCP should be given. Accordingly, we will extend section 1.3 of the introduction with following information:**
*"The first production phase is the spring bloom, which is controlled by the availability of nitrate and shifted from being dominated by diatoms to dinoflagellates in the late 1980s (Wasmund et al., 2017; Spilling et al., 2018). After a so-called blue water period with close–to–zero NCP rates, the second production phase consists of mid–summer blooms dominated by nitrogen-fixing cyanobacteria that develop in most years depending on meteorological conditions. Although cyanobacteria NCP is yet poorly constrained, its relative contribution to the annual NCP in the Eastern Gotland Sea in 2009 was estimated in the order of 40% (Schneider and Müller, 2018; Schneider et al., 2014), though the uncertainty is high. This preliminary estimate further needs to be interpreted with care as cyanobacteria NCP varies significantly between years and regions."*
**This additional information was also requested by RC2.**

In the paper you mention that your approach to NCP (using surface pCO2 and the modelled MLDs/TPDs) is limited to the time-periods of a stable or shoaling thermocline. Would you go as far as to estimate the % of time when the bloom falls outside this pattern based on your

modelled MLD/TPD? This could give you some information/confidence about the applicability of the NCP approach using surface pCO2 only in the lack of full profiles.

**As pointed out in the discussion (L420ff) and in line with several previous studies, we did not observe signs for a continued net production of organic matter during or directly after the deepening of the mixed layer. We therefore conclude that bloom events are entirely confined to periods of a stable or shoaling thermocline. We will explicitly state this by adding in L421:**
*"We thus conclude that reconstructed NCP estimates are not affected by a systematic underestimation due to this temporal restriction."*

The authors may want to include the calculations of heat flux and use it as a proxy for CT* drawdown. These will likely correlate.

**If our understanding of this comment is correct, you are asking for the correlation between heat fluxes across the air-sea interface and the $C_T^*$ drawdown near the sea surface. In first approximation, the heat flux should be strongly correlated with the SST increase over the bloom period and under the condition of a stable thermocline, i.e. from June 6 to 24 in this study. Therefore, we argue that the heat flux can be replaced by SST increase to address this comment.**
**Indeed, previous studies revealed a strong positive correlation between the $C_T^*$ drawdown and the increase in SST (rather than absolute SST) for individual bloom events (Schneider and Müller, 2018; and references therein). Although a strong correlation exists, the slope of this relationship revealed significant variability in space and time. As a consequence, the $C_T^*$ consumption cannot be accurately predicted from a known SST increase or heat flux. The observational constraint of the $C_T^*$ consumption therefore remains an essential component to determine NCP. In order to avoid the potential misconception that the $C_T^*$ consumption can be constrained when only the heat flux or SST is known, and because we see no added value in re-addressing the previously discussed correlation, we do not intend to include the correlation between heat fluxes and $C_T^*$ drawdown in this study. However, the previously found correlation between $C_T^*$ drawdown and SST increase will be addressed as follows in Sect. 1.5 of the introduction:**
*"... it was found that the $C_T^*$ drawdown correlates well with the co–occurring increase in sea surface temperature (SST), rather than with absolute SST. This relationship was attributed to a common driver, which is the light dose received by the water mass under consideration (Schneider and Müller, 2018)."*

Specific comments:

L15: specify units of NCP (mol m^-2 (?))

*"in moles of carbon per surface area"* **will be inserted next to the first mention of NCP in L5.**

L20: within 10% (if taking into account the period of a stable thermocline?) please specify
**For clarification, "*... over the productive period*" will be added to the sentence. We consider the duration of the productive period to be identical to the period of a stable thermocline. This coincidence was further clarified in L421. (Please refer also to our reply to the respective major comment.)**

L30: see the general comments re conventional definition of NCP. Please add the missing info here.

**This will be done. Please refer to our answer to your respective general comment.**

L33: 1.2.Baltic Sea: please add more references citing studies of drivers and causes of hypoxia/anoxia in the Baltic Sea and sealed estuaries in general.

**The following additional information and references were added to support the readers access to the most relevant literature:**
**"*The export of organic matter into the deep waters is considered the ultimate cause for the expansion of anoxic areas in the Baltic Sea, which are nowadays among the largest anthropogenically induced anoxic areas in the world (Carstensen et al., 2014). Although the actual oxygenation state of the deep basins of the Baltic Sea is modulated by the frequency and strength of inflow events (Mohrholz et al., 2015; Neumann et al., 2017) and the biogeochemical properties of the inflowing waters (Meier et al., 2018), the long–term expansion of the anoxic water body was primarily attributed to increased nutrient inputs from land (Jokinen et al., 2018; Meier et al., 2019; Carstensen et al., 2014; Mohrholz, 2018) that fueled the organic matter production in surface waters.*"**
**This additional information and the paraphrasing of the original quote taken from *Carstensen et al. (2014)* was also requested by RC2.**

L47: how important is the spring bloom in the Baltic Sea relative to the summer bloom? Which species dominate during the spring bloom?

**Both questions will be addressed. Please refer to our answer to your respective general comment.**

L58: 'The limited understanding of the factors...' This study does not explicitly address the factors controlling the bloom, instead examines the ways to quantify NCP. This paragraph needs rewording to include background information relevant to this study.

**This issue was also raised by RC2 and we agree that this study itself does not explicitly address the controlling factors of the blooms. However, we expect a major contribution to this question when applying the new NCP reconstruction approach to almost two decades of SOOP observations. In order to clarify this, we will add the following sentence in L62:**
**"*A long–term hindcast of cyanobacteria NCP and the attribution of its strength to prevailing environmental conditions in particular years could improve our understanding of controlling factors and facilitate more reliable predictions of the blooms. However, such a hindcast of cyanobacteria NCP was so far impossible due to**

*missing vertically-resolved observations that would allow to constrain their organic matter production."*
**Please refer also to our answer to your respective general comment.**

L73: cite the literature examining equilibration time of gases.

**The following citation will be included: Wanninkhof (2014)**

L75-L80: do these statements refer to the Baltic Sea? Need to make this clear to the reader. Global N-based NCP estimates are very popular)

**The statement refers to the general limitation that NCP can not be determined from the consumption (i.e. decreasing concentration) of nitrate for blooms of N-fixing organisms, which also applies to Baltic Sea cyanobacteria blooms. To clarify this, the respective sentences will be rephrased to:**
*"However, time series of nutrient consumption do not allow for determining NCP of algae blooms dominated by nitrogen-fixing organisms and those with highly variable C:P ratios. As both characteristics are typical for Baltic Sea cyanobacteria blooms (Nausch et al., 2009), the well established $C_T$ approach is the favorable method to determine mid-summer NCP in this region."*

L136: what are the accuracies and precision of the sensors after their calibrations?

**The sentence in L136 will be rephrased as follows to address the measurement quality of the CTD instrument:**
*"Pre- and post-deployment calibrations of the instrument were carried out in the accredited calibration lab of the IOW in the time span of a few month around the deployments and confirmed that the temperature and conductivity sensors achieved the typical accuracy of better than ±0.01 °C and ±0.01 S m$^{-1}$, respectively."*

L148: 'of 1% of reading'. Is this true for this particular study? Did you compare pCO2 (sensor) to pCO2 (DIC-TA)? If so, this comparison should be included instead.

**To address the sensor performance more clearly in the methods section, we will move the following information from the appendix to L148:**
*"Given the statistics of the pre– and post–deployment calibration, the small drift encountered throughout the deployment and the otherwise smooth performance and regular cleaning of the sensor during the deployment, the accuracy of the measurements is considered to be within 1% of reading as also found by Fietzek et al. (2014)."*
**Further details illustrating the excellent sensor performance are given in Appendices A1 – A3.**
**Unfortunately the suggested comparison of measured $pCO_2$ and $pCO_2$ calculated from DIC and TA is not meaningful to address the uncertainty of the measured values in the Baltic Sea. It was previously shown that calculated $pCO_2$ can deviate from measured $pCO_2$ by more than 200 μatm (Kuliński et al., 2014), due to contributions of dissolved organic acids to TA, which are not accounted for in routine $CO_2$ system calculations.**

**We avoid a direct comparison, because the accuracy of calculated pCO$_2$ values is far lower than the expected accuracy of measured pCO$_2$.**

L149: Please justify using A4.1 for tau determination and A4.2 for the correction? Why not to stick to one equation? Both work great, although A4.2 add more noise to the data.

**We agree that our equation A4.2 can be replaced by A4.1 solved for the in situ pCO$_2$ (i.e. the true value), as originally proposed by (Miloshevich et al., 2004) and previously applied to measurements performed with the HydroC pCO$_2$ sensor by Fiedler et al. (2013) and Atamanchuk et al. (2015). Comparing our original response time corrected pCO$_2$ values based on A4.2 with those calculated based on the rearranged version of A4.1, we found that the vertical profiles gridded to 1m depth intervals are identical, i.e. the choice of the equation used has no impact on the biogeochemical interpretation of our measurements. However, we agree that there is no reason to apply two different types of equations and therefore replace the previous A4.2 with the rearranged version of A4.1 in our calculations and the appendix.**

L159: based on (the length) of the release line??

**Yes. The sentence was rephrased accordingly.**

L160: HgCl2

**Corrected.**

2.4.1 should go before L 176

**The order of the sections was changed.**

L177: in my opinion, the term 'estimate' already refers to an indirect, non-empirical way to assign value to a certain phenomenon. 'Best-guess' should, therefore, be dropped.

**We agree that our combined use of the terms "best-guess" and "estimate" represents a pleonasm. Still, we deem it very important to clearly distinguish our two types of estimates, i.e. the "best–guess" based on vertically resolved observations and the "reconstruction" based on surface observations. As dropping the term "best–guess" would reduce clarity, we will drop "estimate" when used in conjunction with "best-guess". For example in L6 *"... providing a best–guess NCP estimate ..."* will be replaced with *"... providing a NCP best–guess ..."*.**

L276: 'Since July 6th ….' what portion or % of the production you expect prior to July 6th?

**Due to the availability of SOOP observation before July 6, it is indeed possible to approximate the production of this bloom that occurred before the first sampling event of the field campaign. The following information will be added to the description of the SOOP observations in L348ff:**
*"Based on SOOP observations before July 6, first signs of the onset of the investigated bloom event were detected on July 3. Between July 3 and 6, an SST*

*increase of ~1 °C was accompanied by a $C_T$\* drawdown of ~10 µmol kg$^{-1}$ (data not shown). Still, in the absence of any vertically resolved observation for this time period, the following comparison of the reconstructions to the best-guess needs to be restricted to the period July 6 – 24 during which the bulk of NCP occurred."*

L293: 'that NCP has ceased' …do you mean there is no production (NPP), nutrient limitation or that NPP is balanced by respiration?

**As the $C_T$\* approach resolves only the net effect of production minus respiration of organic matter, this statement refers to the fact that production is roughly balanced by respiration. This will be clarified by rephrasing the sentence to:**
*"... indicating that the production and respiration of organic matter were balanced during this period"*

Fig. 5
f)-g) Fair-sea or Fatm for clarify on Y-axis
h) flux = air-sea flux. 'Flux' is too ambiguous in this case. flux mixing corrected ( -NCP ) – production has a positive sign, so negative to integrated CT\*

**The labels will be changed as you recommend, both in figures and the corresponding manuscript text.**

Fig.6.
c) again, NCP must be positive by conventional definition. Either use NCP(CT) or some other way to clarify why Net Community Production in your case is negative.

**The sign of NCP will be clarified. We will use "- NCP" in the figure in order to ensure the compatibility of $C_T$\* and NCP time series plots. In the manuscript text, we will point out that the decrease of $C_T$\* corresponds to an increase (opposite sign) of NCP. In agreement with a comment by RC2, we will also include our equation used to calculate NCP, which will further help to clarify signs.**

L400: Please also explain how to overcome the caveats. This info can be useful for the reader. What other measurements are needed?

**In the new section (L399) we will include a discussion of the biogeochemical relevance and interpretation of our findings, where we will address this question as follows:**
*"We conclude that NCP estimates determined with the methods developed in this study are of direct relevance to quantify the drivers for deep water deoxygenation. However, a better understanding of the organic matter remineralisation processes would be required to close the budget of biogeochemical transformations. New observational platforms, such as recently deployed biogeochemical ARGO floats (Haavisto et al., 2018), will complement the existing SOOP infrastructure and help to provide the required observational constraints throughout the water column."*

L420: Indeed, it is unlikely that production will continue during the periods of deepening thermocline. However, deepening-restratification events may have an impact on both

entrainment of CT and detrainment of POC. The latter, especially, can moderate more efficient transport from the photic zone to the depth bypassing the respiration stage. Therefore, it appears crucial to monitor the frequency of occurrence of such events as they impact both the production and efficiency of carbon export.

**In the new section (L399) we will include a discussion of the biogeochemical relevance and interpretation of our findings, where we will address this question as follows:**
*"In contrast to shallow remineralisation processes, the deepening of the mixed layer that marked the end of the studied bloom event may facilitate the efficient transport of POC from the surface layer to depth. Focusing on the accumulation of remineralisation products beneath 150 m in the Gotland basin, a previous study revealed that – in accordance with the main input of POC during the productive period – remineralisation rates exhibit a pronounced seasonality (Schneider et al., 2010). This seasonality was found to be most pronounced in the water layers closest to the sediment surface, suggesting that beneath 150 m the remineralisation takes place mainly at the sediment surface and is of minor importance during particle sinking through the deep water column. The pronounced seasonality further confirms that surface organic matter production and deep water oxygen consumption are indeed tightly coupled, despite a potential degradation of POC before export across the permanent halocline."*
**This additional information was also requested by RC2.**

**Additional references used in this reply**

Atamanchuk, D., Tengberg, A., Aleynik, D., Fietzek, P., Shitashima, K., Lichtschlag, A., Hall, P. O. J., and Stahl, H.: Detection of CO2 leakage from a simulated sub-seabed storage site using three different types of pCO2 sensors, Int. J. Greenh. Gas Control, 38, 121–134, https://doi.org/10.1016/j.ijggc.2014.10.021, 2015.

Fiedler, B., Fietzek, P., Vieira, N., Silva, P., Bittig, H. C., and Körtzinger, A.: In Situ CO2 and O2 Measurements on a Profiling Float, J. Atmospheric Ocean. Technol., 30, 112–126, https://doi.org/10.1175/JTECH-D-12-00043.1, 2013.

Kuliński, K., Schneider, B., Hammer, K., Machulik, U., and Schulz-Bull, D.: The influence of dissolved organic matter on the acid–base system of the Baltic Sea, J. Mar. Syst., 132, 106–115, https://doi.org/10.1016/j.jmarsys.2014.01.011, 2014.

Miloshevich, L. M., Paukkunen, A., Mel, H. V., and Oltmans, S. J.: Development and Validation of a Time-Lag Correction for Vaisala Radiosonde Humidity Measurements, J. ATMOSPHERIC Ocean. Technol., 21, 24, 2004.

Nausch, M., Nausch, G., Lass, H. U., Mohrholz, V., Nagel, K., Siegel, H., and Wasmund, N.: Phosphorus input by upwelling in the eastern Gotland Basin (Baltic Sea) in summer and its effects on filamentous cyanobacteria, Estuar. Coast. Shelf Sci., 83, 434–442, https://doi.org/10.1016/j.ecss.2009.04.031, 2009.

Sarmiento, J. L. and Gruber, N.: Ocean Biogeochemical Dynamics, Princeton University Press, 528 pp., 2006.

Schneider, B. and Müller, J. D.: Biogeochemical Transformations in the Baltic Sea, Springer International Publishing, Cham, https://doi.org/10.1007/978-3-319-61699-5, 2018.

Wanninkhof, R.: Relationship between wind speed and gas exchange over the ocean revisited: Gas exchange and wind speed over the ocean, Limnol. Oceanogr. Methods, 12, 351–362, https://doi.org/10.4319/lom.2014.12.351, 2014.

**AC2: Reply on RC2**

Provided by Jens Daniel Müller on behalf of all co-authors

**Dear Referee 2,**

**Thank you for providing your review, which we considered very helpful to strengthen the presentation of our study. Most of your comments request an extension of the manuscript with additional information, which we are happy to implement. In particular, we aim to add a dedicated discussion section addressing the biogeochemical interpretation of our NCP estimates. Several points you raised were also in agreement with RC1, and we cross reference our answers where this applies.**

**Please find our detailed answers (bold font) and proposed text edits (bold italic font) next to your comments (normal font) below. Line numbers refer to the initially submitted version of the manuscript.**

**We hope to have addressed all of your comments appropriately, but welcome additional feedback if required.**

**Best wishes**
**Jens Daniel Müller, on behalf of all co-authors**

Specific comments:

Line 46 – It would be interesting and relevant to the study background if the authors could clarify what is the relative contribution of the second, summertime cyanobacteria to the Baltic Sea hypoxia?

**Similar information was also requested in RC1. To our understanding, this question involves two aspects: (1) What is the contribution of cyanobacteria to the annual NCP including the spring bloom and (2) how does NCP relate to deep water deoxygenation.**

**We intend to address aspect (1) as follows:**
**Enabling reliable NCP estimates for the mid-summer cyanobacteria bloom is the core aim of this study. A reliable hindcast of the mid-summer NCP will only be possible when the findings of this study are applied to almost two decades of available SOOP pCO$_2$ data. In the absence of this information, the assessment of the importance of the spring relative to the mid-summer bloom is highly uncertain. Nevertheless, we agree that some more information on the spring-bloom and a rough approximation of its contribution to the annual NCP should be given. Accordingly, we will extend section 1.3 of the introduction with following information:**
***"The first production phase is the spring bloom, which is controlled by the availability of nitrate and shifted from being dominated by diatoms to dinoflagellates in the late 1980s (Wasmund et al., 2017; Spilling et al., 2018). After a so-called blue water period***

*with close–to–zero NCP rates, the second production phase consists of mid–summer blooms dominated by nitrogen-fixing cyanobacteria that develop in most years depending on meteorological conditions. Although cyanobacteria NCP is yet poorly constrained, its relative contribution to the annual NCP in the Eastern Gotland Sea in 2009 was estimated in the order of 40% (Schneider and Müller, 2018; Schneider et al., 2014), though the uncertainty is high. This preliminary estimate further needs to be interpreted with care as cyanobacteria NCP varies significantly between years and regions.”*

We intend to address aspect (2) as follows:
We will add a dedicated section in L399 of the discussion to describe the biogeochemical relevance and interpretation of our NCP estimates. Among others, this section will address the general relation between NCP, organic matter export and deoxygenation as follows:

*““Our best–guess of cumulative NCP on July 24 (~1.2 mol m$^{-2}$) represents the net amount of organic matter that was produced throughout the bloom event in the surface waters above the compensation depth at 12 m. After subtracting ~20 % dissolved organic carbon (DOC) production, our NCP estimate equals the produced particulate organic carbon (POC) that is potentially available for export. [...] However, the potential POC export constraint by our NCP estimate is not equivalent to the supply of organic matter to the deep waters of the Gotland Basin, because POC might be (partly) remineralised before sinking beneath the permanent halocline. Remineralisation of POC that occurs during the bloom event above the compensation depth is – according to our definition of NCP – already included in our estimate. In contrast, any additional remineralisation of POC that occurs between the compensation depth and the halocline, or above the compensation depth after the end of the bloom event, reduces the organic matter supply to the deep waters and thereby mitigates deoxygenation. Indeed, our profiling measurements indicate a steady accumulation of $C_T$\* beneath the compensation depth (Fig. 4), likely fueled by the remineralisation of organic matter. However, our measurements do neither allow to constrain the budget of this $C_T$\* accumulation, nor could we attribute the source of organic matter.”*

Section 1.3 – It is not very clear how this study will “disentangle” the multiple stressors on cyanobacteria blooms in the Baltic Sea. The way this section is written promises the reader that the results and discussion will address this challenge, even though it does not clearly do so. I would instead focus the background here more on the relationship among cyanobacteria blooms, hypoxia and NCP, which better relates to the study's aims.

**This issue was also raised by RC1 and we agree that this study itself does not explicitly address the controlling factors of the blooms. However, we expect a major contribution to this question when applying the new NCP reconstruction approach to almost two decades of SOOP observations. In order to clarify this, we will add the following sentence in L62:**
*“A long–term hindcast of cyanobacteria NCP and the attribution of its strength to prevailing environmental conditions in particular years could improve our understanding of controlling factors and facilitate more reliable predictions of the blooms. However, such a hindcast of cyanobacteria NCP was so far impossible due to*

*missing vertically-resolved observations that would allow to constrain their organic matter production."*

**We will further clarify how hindcasts based on our findings will support the disentangling of drivers by adding the following information to our conclusions (L437):**
*"The application of this approach will allow for the detection and attribution of trends in cyanobacteria NCP across decades. In particular the comparison of NCP estimates of bloom events that occurred under different environmental conditions will provide a better understanding of the controlling factors. Factors to be tested include the environmental parameters used to constrain NCP ($pCO_2$, SST, and TPD), but also additional observations of nutrients and phytoplankton composition routinely determined on SOOP Finnmaid and in the framework of the Baltic Sea monitoring program. The recently started initiative to deploy biogeochemical ARGO floats in the Baltic Sea will further aid to link surface NCP estimates and deep water deoxygenation, and thereby constrain biogeochemical budgets in the Baltic Sea."*

Line 71-73 – It is not clear, in the context of this study, why using oxygen measurements to estimate NCP is inferior to using $pCO_2$ to estimate NCP, except that $pCO_2$ data are perhaps more readily available from ships of opportunity. There are multiple examples of NCP being estimated using dissolved oxygen time series (e.g., â^†$O_2$/Ar time series) on the time scales of phytoplankton blooms. If $O_2$ equilibrates more quickly than $pCO_2$, would it not be preferable for studying Baltic Sea dynamics over a few weeks? In any case, in this study, the authors report cumulative NCP estimates over time, which suggests that, despite the different equilibration time scales for $O_2$ and $CO_2$ in the mixed layer, cumulative NCP estimates based on each parameter should approximate each other over the time scale of a bloom.

**We think that an essential piece of information was missing to make our statement misunderstandable. NCP estimates based on either $CO_2$ or $O_2$ time series require a correction of the observed concentration changes for the air-sea flux of either gas. The calculation of this air-sea flux is associated with uncertainties. As a consequence, the higher flux rates of $O_2$ lead to a higher uncertainty in the derived NCP estimate. We will try to clarify this by modifying Line 71-73 as follows:**
*"In principle, NCP could as well be estimated from $O_2$ time series. However, the equilibrium reactions of carbon dioxide ($CO_2$) in seawater result in slower re–equilibration of $CO_2$ with the atmosphere compared to $O_2$ (Wanninkhof, 2014). This results in substantially longer preservation of the $C_T$ signal and a lower uncertainty contribution of required air–sea $CO_2$ flux corrections, and makes $C_T$ the preferred tracer for NCP."*

Section 2.2.3 – It would be useful here to state the different phytoplankton that were collected by name, rather than in Appendix B2 only.

**This information will be included in Line 164 as follows:**
*"Phytoplankton samples were fixed with Lugol solution, and cyanobacteria community composition and biomass were determined by microscopic counts of the*

*genera Aphanizomenon, Dolichospermum and Nodularia according to the Utermöhl method (HELCOM, 2017)"*

Section 2.4 – It would be very helpful to provide the equations used to calculate the "best-guess" NCP values, including how the integration depth was determined in this approach. In the results, the authors imply that they used a depth of 12 m (lines 308-310), but this should be explained in the Methods rather than in the results.

**In agreement with a comment from R1, we will clarify that our determination of the integration depth aligns with the traditional concept of the compensation depth, i.e. the depth at which primary production and respiration balance out. Accordingly, we will clarify in Line 180 that our NCP best-guess is constrained to the compensation depth:**
*"... we derive the column inventory of incremental changes of $\Delta C_T^*$ ($i\Delta C_T^*$) between two cruise events through vertical integration of $\Delta C_T^*$ from the sea surface to the compensation depth (cd), i.e. the depth (z) at which no net drawdown of $CO_2$ was observed"*
**This clarification in the methods section will further be supported by two equations summarizing our NCP calculation.**

Lines 184-191 – It would be useful to clarify here that applying an average alkalinity to derive Ct* is valid because, as the results will show, the biogeochemical variability across stations of interest was low.

**The information given in lines 190 - 191 was rephrased and now reads:**
**"*The uncertainty in the determination of changes of $C_T^*$ is below 2 µmol kg⁻1 when the mean $A_T$ is constrained within the observed standard deviation of ± 27 µmol kg$^{-1}$ (see Appendix C1 for a detailed assessment).*"**

Section 2.4.4. – This section about vertical mixing should reference Figure 3 because the vertical mixing event is very clear when looking at all the profiles at once. Referencing Figure C3 seems redundant (see my technical comment below).

**The reference to Figure 3 will be included in line 208. (However, there was no redundant reference to Figure C3 in section 2.4.4. that could be removed)**

Line 249 – Is it fair to say that, because the BloomSail data exhibit little regional biogeochemical and physical variability across the stations of interest, the cruise tracks encompassing this same region should not be variable as well? If so, perhaps the authors could clarify that here, as well.

**This is absolutely fair to say. It will be clarified in line 250 that the regional variability of averaged physical parameters from the GETM model was low.**

Lines 259-265 - While the general idea is there, the authors need to better explain how they obtain TPD values. It was unclear in this paragraph and in Fig. C4A how they choose an actual depth. Is there a threshold for change in temperature between cruises that helps one

select the right depth (e.g., the depth when change in temperature is 0.2 °C, according to Fig. C4A?), analogous to using a density difference criterion for deriving mixed layer depth?

**No, for TPD no temperature threshold is required. We will try to clarify this and the approach in general by rephrasing the respective section to:**
**"*TPD characterises the mean penetration depth of surface warming that occurred between two sampling events. TPD was defined as the SST increase divided by the integrated warming signal across the water column, i.e. the sum of all positive temperature changes within 1m depth intervals (for a graphical illustration see Fig. C4A). According to this definition and in contrast to MLD, TPD takes gradual changes of temperature across depth into account and does not require a fixed threshold value. TPD is only applicable when SST increases and has units of metres. To illustrate the TPD concept, it should be noted that a homogeneous warming signal that ceases abruptly at 10 m water depth would result in the same TPD as a warming signal that decreases linearly from the surface to 20 m water depth (TPD is 10 m in both cases)*"**

Lines 308-310 – How does this choice of integration depth compare to a best-guess NCP estimate calculated using a mixed layer depth based on a density difference criterion? Should the "best-guess" NCP estimate also be calculated using this density difference approach so that it is more comparable to the reconstructed NCP calculations later on in the paper?

**We did actually calculate NCP based on a density difference criterion (MLD) and results are displayed in Fig. 6c (left panel). One could argue that this estimate is based on surface $CO_2$ observations only, and is therefore a reconstruction and not a best-guess. However, the vertical variability of the $C_T$\* profiles above the MLD is very small (compare Fig 4, a2) and accordingly there is no significant difference between integrating the surface values or vertically resolved $C_T$\* values across the MLD. We therefore argue that the requested comparison is already covered in Fig. 6c.**

Figure 5 – I am used to reporting NCP as positive if Ct\* decreases, and negative if Ct\* increases. Thus, I do not understand how NCP was negative, until August 6, and positive from August 6-13, unless the authors are using an opposite sign? (I would expect the opposite because inorganic carbon drawdown would indicate more production over respiration.) I suggest the readers reevaluate the sign of the NCP values they report here. This is another reason that it would be important for them to share their NCP calculation equations in Section 2, as well.

**This comment is in agreement with a remark by R1. To clarify the issue we will indicate in Figure 5 and the corresponding caption, that the sign of NCP is indeed the opposite of the changes in $C_T$\*. We will also include two equations for NCP calculation in Sect. 2 and explicitly mention the interpretation of the sign of the three components (observed $C_T$\* changes, air-sea fluxes, and mixing).**

Lines 321-322 – It would be interesting for the authors to discuss this peak cumulative NCP value more in the discussion. How does resolving the change in cumulative NCP over the course of a bloom improve understanding around hypoxia in the Baltic Sea?

**The contribution of cumulative NCP estimates to the better understanding of hypoxia in the Baltic Sea will be addressed in a dedicated section in the discussion. Please refer to our answer to your comment on Line 46 above and also the reply to RC1.**

Lines 323-324 – If the authors focused on reconstructing NCP over July 6 to July 24, why does Figure 6 show reconstructed values beyond July 24? The figure is not consistent with the text in this respect.

**This sentence was indeed unclear. Our intention was to point out that our presentation of the results and their discussion and interpretation will focus on the period until the NCP peak on July 24, not that the reconstruction approach is technically limited to this period. To avoid this misunderstanding, we will rephrase the respective sentence to:**
**"*Accordingly, our interpretation of the reconstructed NCP based on surface $pCO_2$ observations will focus on the NCP peak value on July 24.*"**

Section 4.1 – Since different NCP units are being reported throughout this section, the authors should make sure to always translate units explicitly in the text to show how they are comparable. For example, how would the authors convert their NCP units to be directly comparable to the values from Wasmund et al. (2001)?

**We already pointed out the that mean rate of 5 µmol kg$^{-1}$ d$^{-1}$ determined in this study refers to the average $C_T^*$ drawdown of ~90 µmol kg$^{-1}$ over 18 days (Line 377). However, in between our initial response to your review and the resubmission of the manuscript we had to correct the given duration from 12 (wrong number given in original submission) to 18 days (correct value). We will further clarify how we compared our mean rate with the values from Wasmund et al. (2001):**
**"*Wasmund et al. (2001) conducted $^{14}C$ incubation experiments at different water depths to determine instantaneous rates of daytime primary production during a cyanobacteria bloom. Their reported carbon fixation rates in surface waters (0.4 – 0.8 mmol C m$^{-3}$ h$^{-1}$) are in the same order of magnitude as the mean rate found in this study (5 µmol kg$^{-1}$ d$^{-1}$, equivalent to 0.2 mmol C m$^{-3}$ h$^{-1}$), despite representing daytime production rates and diurnal averages, respectively.*"**

**However, it should also be emphasized that the comparison to the findings of Wasmund et al. (2001) focuses rather on the depth distribution of NCP rather than the absolute values, for which the unit conversion is irrelevant. This aspect will be highlighted as:**
**"*More important than the agreement between the fixation rates at the sea surface, is the fact that Wasmund et al. (2001) also found significantly lower fixation rates below 10 m water depth (<0.2 mmol–C m$^{-3}$ h$^{-1}$), which agrees well with the depth distribution of NCP observed in this study.*"**

Section 4.2 - Scaling this study's method depends on the availability of models like the GETM for other locations. Can the authors comment on the availability and applicability of this model (or other similar models) in other ocean regions?

**We will clarify in the methods sections that the model run we used**
**"... covers the entire Baltic Sea and the period 1961 - 2019."**

**With respect to the applicability outside the Baltic Sea, it should be noted that in general GETM is a state of the art ocean model as it is ROMS or NEMO. As long as good enough forcing data are available (atmospheric data, boundary data) and the models spatial and vertical resolution is sufficient to resolve the important scales, we do not see any show stopper in applying GETM (or other models) to other regions of the coastal or global ocean. At present GETM is used for studies in the North Sea / Celtic Sea, the Mediterranean Sea, the Black Sea, Persian Gulf, and along the west coast of Australia. However, we deem it beyond the scope of this study to provide such detailed information on the availability/applicability of GETM or other ocean models. We would even be afraid that too much information to this end distracts from the core findings of this study or creates the impression that we consider our reconstruction approach ready-to-go for other ocean regions. Therefore, we prefer to refrain from adding this information to the manuscript.**

Line 405 – That a mean regional alkalinity could be used to normalize dissolved inorganic carbon seems to be an important output of the study. Even though this is not the main objective, perhaps this outcome should be acknowledged as a goal in the introduction, as well.

**Although this study confirms that a mean regional alkalinity can be used to quantify changes in normalized dissolved inorganic carbon concentration over time and provides a detailed uncertainty assessment for the underlying conversion from $pCO_2$, this approach is not a new outcome or intellectual achievement of this study. In contrast, we rely on this previously established method, which is explicitly stated in the introduction as:**
**"... it was demonstrated that highly accurate time series of changes (not absolute values) in $C_T$ can be derived from $pCO_2$ observations (Schneider et al., 2006). The conversion from $pCO_2$ to $C_T$ relies on a fixed alkalinity ($A_T$) estimate and is applicable under the condition that internal sources of $A_T$ can be excluded, which is the case in the Baltic Sea due to the absence of calcifying plankton (Tyrrell et al., 2008)."**
**and:**
**"This study builds upon the previous success to determine NCP based on $pCO_2$ time series, but extends the approach to vertically resolved observations for the first time."**
**Accordingly, we do not intend to introduce this outcome as a goal of this study.**

Technical comments:

Line 39-40 – It is better to paraphrase this quotation than quote it directly from the source.
**The quotation will be paraphrased.**

Lines 216-217 and 220-222 – I suggest writing out these bullet points into full sentences to be stylistically consistent with the rest of the manuscript.

**We appreciate this suggestion, but feel that the bullet points help to present the various reconstruction approaches in a structured fashion. Furthermore, we do not believe that this is stylistically inconsistent with the rest of the manuscript, as there are no similar lists of approaches, which are not set as bullet points. Therefore, we prefer to keep the formatting as is.**

Figure 2 – The authors do not need color here to convey the date. Mean cruise date could just be written into the x-axis labels (with labels rotated so that the text fits).

**We agree that in Figure 2 itself, using color to convey mean cruise dates is somewhat redundant with the date on the x-axis. However, the intention of using color in Figure 2 is to directly link it to Figures 3, 4, and B1, in which we use the exact same color scale. Furthermore, writing mean cruise dates directly on the x-axis would result in unequal breaks of the time axis, which we want to avoid. Therefore, we intend to keep Figure 2 as is.**

Line 330-331 – This sentence, "For both data sets CT* time series were calculated based on the same mean AT", should go in the methods, Section 2.5.

**The sentence will be relocated.**

Line 359-361 – This sentence is grammatically confusing and should be rephrased.

**We agree that this sentence was confusing and combined two statements, each of which deserves its own sentence. The sentence will be split and rephrased to:**
**"*Still, this lateral variability is small compared to the signal to be resolved (i.e. the $C_T$\* drawdown of ~90 µmol kg$^{-1}$). However, on a relative scale the lateral $C_T$\* variability is about as large as the difference between the best–guess and the TPD–based NCP reconstruction (~10%), suggesting that the bias of the reconstruction falls within the uncertainty range of the best-guess.*"**

Line 421- I suggest using a different word than "planktological".

**"planktological findings" will be replaced with:**
**"*findings from the long-term cyanobacteria monitoring program*"**

Figure C3 – This is not necessary because the same information is conveyed in Figure 3 (therefore this figure seems redundant).

**While it is true that the same data are shown in Figure 3 and C3, the information that should be conveyed differs. While Figure 3 is intended as a first, compact overview relating all measured profiles to each other on one plot, Figure C3 seeks to resolve the variability across stations on one cruise day. This higher resolution is critical to underpin some of our statements in the results section (e.g. "During this period of intense primary production, the regional variability of SST, pCO$_2$, and $C_T$\* across stations was low compared to their temporal change") as well as in the discussion (e.g. "The temporary $C_T$\* increase was limited to the north–eastern stations 07 – 10"). We therefore intend to keep both figures as they are.**

**Additional references used in this reply**

Spilling, K., Olli, K., Lehtoranta, J., Kremp, A., Tedesco, L., Tamelander, T., Klais, R., Peltonen, H., and Tamminen, T.: Shifting Diatom—Dinoflagellate Dominance During Spring Bloom in the Baltic Sea and its Potential Effects on Biogeochemical Cycling, Front. Mar. Sci., 5, https://doi.org/10.3389/fmars.2018.00327, 2018.

Wanninkhof, R.: Relationship between wind speed and gas exchange over the ocean revisited: Gas exchange and wind speed over the ocean, Limnol. Oceanogr. Methods, 12, 351–362, https://doi.org/10.4319/lom.2014.12.351, 2014.

Wasmund, N., Kownacka, J., Göbel, J., Jaanus, A., Johansen, M., Jurgensone, I., Lehtinen, S., and Powilleit, M.: The Diatom/Dinoflagellate Index as an Indicator of Ecosystem Changes in the Baltic Sea 1. Principle and Handling Instruction, Front. Mar. Sci., 4, https://doi.org/10.3389/fmars.2017.00022, 2017.

---

## Referee Report (RR1)

Dear Dr. Marañón,

I find that the manuscript, "Cyanobacteria net community production in the Baltic Sea as inferred from profiling pCO2 measurements", has improved greatly since its original submission. The authors have made effort to clarify the unique contributions that their work has for understanding primary productivity patterns in the Baltic Sea. After reading this second submission, it is now clear to me how their approach to reconstructing NCP from modeled temperature/salinity profiles and opportunistic $pCO_2$ measurements could improve understanding of eutrophication and hypoxia in this marine environment. In particular, I am impressed by the potential opportunity to reconstruct NCP over past decades using surface $pCO_2$ measurements from ships of opportunity, and I hope that the authors pursue this research endeavor themselves or help other researchers in the region do so.

Perhaps most importantly, the authors have improved the methods section of the manuscript, providing a thorough explanation of their different approaches to calculating depth- and time-integrated NCP values. I think it is now possible for an external reader to replicate and apply these methods in other contexts.

Overall, I recommend that this manuscript be accepted for publication with minor revisions. The comments in the following letter address some further suggestions and questions I have about the methods and results that I think should be addressed before publication. My comments are separated by section and/or line number, as relevant.

Sect. 2.2.3: How were the $C_T*$ values from discrete measurements used in this study? I suggest that the authors clarify the way these discrete measurements were used. If they were just used for comparison to $C_T*$ values derived from $pCO_2$ profiles, what were the results of those comparisons?

Section 2.5: It is important to reference Section 2.6.3 in Section 2.5, which explains the calculation of the $C_T*$ drawdown penetration depth.

Section 2.6: For thoroughness, I suggest that the authors write out the calculations for reconstructed NCP, as in Section 2.5. Even though they are similar to the previous set of NCP calculations, it would be useful and complementary to see them written out.

Figure C4: This is so useful for understanding Section 2.6.3. Therefore, I think this figure should be in the main text.

Line 147: Does "below" in this sentence means "less than" 60 m depth? I think the authors should clarify this.

Line 185: Why were discrete samples collected at just two stations?

Line 299: I double-checked the math described in this paragraph (Section 2.6.3), and I think there is a typo here. I think TPD should be defined as the integrated warming signal divided by

the SST increase, instead of the other way around (which is how the sentence is written). That's the only way the example with 10 m provided on line 304 would make sense.

Lines 311-312: Similarly, as in my prior comment, I think CPD is the integrated loss of $C_T^*$ divided by the decrease in $C_T^*$ at the surface, rather than the other way around, as it is currently written.

Figure 4: I have a number of comments about this figure. First, it is difficult to see the August 16 data (white circles) for panels a1 and b1, so I suggest extending the x-axis on these two sub-plots. In panels a2 and b2, why are there eight vertical profiles for $\Delta$temperature and $\Delta C_T^*$? If these values indicate changes between cruise events, there should be seven values rather than eight. It is a bit misleading to plot the July 6 profiles, which have values of 0 °C and ~μmol kg$^{-1}$ across depths, as no magnitudes of change could calculated for this first cruise. Finally, why is there just one depth indicated on this figure if the authors allowed the CPD, MLD or TPD values to change throughout the duration of the 8 BloomSail cruises? This contrasts with Figure 6, which indicates variable integration depths.

Line 331: If lateral exchange was important at the northeastern stations, how much did this observed increase in $A_T$ and $C_T^*$ impact the best-guess NCP estimates around the July 31 cruise?

Line 451: I recommend the authors cite again here where they acquired the 20% estimate for DOC production?

Line 483: How does the requirement of a mean measured $A_T$ value for the region of study weaken the utility of this surface- and model-based NCP reconstruction approach, considering that $A_T$ is not measured on ships of opportunity?

---

## Author Response (AR3)

**AC2.2: Reply on RC2.2 (second revision)**

Provided by Jens Daniel Müller on behalf of all co-authors

**Dear Referee 2,**

**Thank you for taking your time to provide a second careful review of our study, after the comments on the original submission were taken into account. The author team is glad that our previous edits helped to clarify the main scientific progress of our study. We also appreciate your encouragement to apply our NCP reconstruction approach to almost two decades of surface pCO2 observations. This is of course what we aim for.**

**Regarding your second revision of our manuscript, the author team considers your suggested minor revisions helpful in order to further clarify methodological details of our study, as well as the interpretation of the results.**

**Please find our detailed answers (bold font) and proposed text edits (bold italic font) next to your comments (normal font) below. Line numbers refer to the resubmitted clean version of the manuscript.**

**We hope to have addressed all of your comments appropriately, but welcome additional feedback if required.**

**Best wishes**
**Jens Daniel Müller, on behalf of all co-authors**

Specific comments:

Sect. 2.2.3: How were the $C_T$* values from discrete measurements used in this study? I suggest that the authors clarify the way these discrete measurements were used. If they were just used for comparison to $C_T$* values derived from $pCO_2$ profiles, what were the results of those comparisons?

**$C_T$* directly measured on discrete samples was indeed only used for comparison to $C_T$* calculated from $pCO_2$. We clarified this use of the data in Sect 2.2.3 (l.191) by adding:**

***The mean observed $A_T$ was used for the calculation of $C_T$* from $pCO_2$ (see Sect. 2.5.2), while measured $C_T$ was only used for comparison to calculated values and not directly included in the NCP calculation.***

**The outcome of this comparison is illustrated in Fig. 5 showing surface $C_T$* from discrete samples along with $C_T$* calculated from $pCO_2$. We further evaluate the agreement between both data sources and added following text in l.327:**

*Furthermore, we found that $C_T$\* calculated from $pCO_2$ agreed with $C_T$\* derived from discrete samples within the uncertainty range attributed to regional variability (Fig. 5c).*

Section 2.5: It is important to reference Section 2.6.3 in Section 2.5, which explains the calculation of the $C_T$\* drawdown penetration depth.

**In Sect. 2.5 and 2.6.3 we describe the two fundamental approaches that we applied to calculate NCP, i.e. the best-guess NCP based on complete $C_T$\* profiles and the reconstructed NCP based on surface $C_T$\*. However, the $C_T$\* drawdown penetration depth (CPD) is not required to determine the best-guess as described in Sect. 2.5 and we thus do not agree that a cross-reference is important or helpful here.**

Section 2.6: For thoroughness, I suggest that the authors write out the calculations for reconstructed NCP, as in Section 2.5. Even though they are similar to the previous set of NCP calculations, it would be useful and complementary to see them written out.

**Thank you very much for this suggestion. Indeed, the additional equation should make it much easier to clearly distinguish our best-guess and reconstructed NCP estimate. The requested equation was added in l.250 of Sect. 2.6. In order to clearly distinguish the equations for our two types of NCP calculations, the NCP term was labeled with indices "best-guess" and "reconstruction" in equations (2) and (3).**

Figure C4: This is so useful for understanding Section 2.6.3. Therefore, I think this figure should be in the main text.

**Following the bg author guidelines, we placed figures with direct relevance to the results in the main text and figures that illustrate methods or reveal details of supplementary nature in the appendices. According to this criterion, we see Fig. C4 correctly placed in the appendix. Please note that Fig. 4C is referenced in Sect. 2.6.3 and that the appendices (in contrast to supplementary materials) will be part of the manuscript (i.e. be printed in the same pdf). We thus conclude that the figure can easily be found by the interested reader.**

Line 147: Does "below" in this sentence means "less than" 60 m depth? I think the authors should clarify this.

**Yes, the text was changed to "*less than*".**

Line 185: Why were discrete samples collected at just two stations?

**As the field sampling of this study was performed with a small sailing vessel and only three crew members, it was not possible to collect discrete samples at more than two stations. We explicitly state this constraint by editing the text in l. 185, which now reads:**

*Discrete samples were collected with a manually released Niskin bottle. Water sampling was restricted to stations 07 and 10 (Fig. 1b) due to logistic constraints.*

Line 299: I double-checked the math described in this paragraph (Section 2.6.3), and I think there is a typo here. I think TPD should be defined as the integrated warming signal divided by the SST increase, instead of the other way around (which is how the sentence is written). That's the only way the example with 10 m provided on line 304 would make sense.

**That is correct. Thank you for spotting the typo. The description was corrected.**

Lines 311-312: Similarly, as in my prior comment, I think CPD is the integrated loss of $C_T$* divided by the decrease in $C_T$* at the surface, rather than the other way around, as it is currently written.

**That is correct. Thank you for spotting the typo. The description was corrected.**

Figure 4: I have a number of comments about this figure. First, it is difficult to see the August 16 data (white circles) for panels a1 and b1, so I suggest extending the x-axis on these two sub-plots. In panels a2 and b2, why are there eight vertical profiles for $\Delta$temperature and $\Delta C_T$*? If these values indicate changes between cruise events, there should be seven values rather than eight. It is a bit misleading to plot the July 6 profiles, which have values of 0 °C and ˜µmol kg -1 across depths, as no magnitudes of change could calculated for this first cruise. Finally, why is there just one depth indicated on this figure if the authors allowed the CPD, MLD or TPD values to change throughout the duration of the 8 BloomSail cruises? This contrasts with Figure 6, which indicates variable integration depths.

**In order to make the markers for the first and last cruise event visible in a1 and b1, we switched from a vertical line to "+" symbols. We do not extend the x-axis, because we want to avoid the required temporal extrapolation beyond the period covered by observations.**

**In panels a2 and b2 we removed the July 6 profiles, while keeping the color scale of the remaining profiles consistent with the other figures.**

**The red vertical line indicates the compensation depth (CD) of the $C_T$* drawdown, which we used as a constraint for our NCP best-guess. In contrast to MLD and TPD, which indeed vary over time, we used a constant CD. It is thus correct to draw a single horizontal line in a2 and b2. Please note also the additional information on this topic given in l.349ff:**

***... the compensation depth located at 12 m. The determined compensation depth reflects the maximum penetration depth of the incremental (i.e. between cruise days), as well as the cumulative (i.e. from July 6 – 24), $C_T$* drawdown (Fig. 4).***

Line 331: If lateral exchange was important at the northeastern stations, how much did this observed increase in $A_T$ and $C_T$* impact the best-guess NCP estimates around the July 31 cruise?

**While the northeastern stations are affected by the lateral exchange and show a temporary increase of $C_T$* (see Fig. C3) on July 31, the other stations are not affected and show an almost constant surface $C_T$* compared to the previous and the following cruise day. We already addressed this in l.328ff of the main text, which reads:**

*Between the extremes of pCO₂ and C_T\* (minimum on July 24) and SST (maximum on August 3), a noticeable increase of surface C_T\* was observed on July 31, which was accompanied by a higher regional variability across the station network (Fig. 5a,c). The temporary C_T\* increase was limited to the north–eastern stations 07 – 10 (Fig. C3) and paralleled by a drop in salinity and elevated A_T at the same stations (Fig. B1). It is therefore attributable to the lateral exchange of water masses. All signals of this lateral intrusion vanished within a week. At the other stations (02 – 06 and 11 – 12), no noticeable signs of water mass exchange or C_T\* changes were observed between July 24 and August 3, indicating that the production and respiration of organic matter were balanced during this period.*

**To confirm our interpretation that the production and respiration of organic matter were balanced during this period, we recalculate our NCP time series without the stations affected by lateral exchange of water masses. Our conclusion from this additional analysis were added in l.365:**

*The temporary drop in the NCP best-guess on July 31 is due to the lateral exchange of water masses as described in Sect. 3.1. Deriving the NCP time series without the stations affected by lateral exchange of water masses (07–10) results in an almost identical NCP estimate on July 24, but a reduced drop on July 31 (data not shown). In both cases, no signs of continued NCP were observed after July 24.*

**Please find at the end of this document modified versions of Figs. 4 and 5 produced without the observations made at stations 07-10. The modified figures serve for demonstration purposes only in this reply to the reviewer, but are not intended to be included in the manuscript for publication.**

Line 451: I recommend the authors cite again here where they acquired the 20% estimate for DOC production?

**The references were added.**

Line 483: How does the requirement of a mean measured $A_T$ value for the region of study weaken the utility of this surface- and model-based NCP reconstruction approach, considering that $A_T$ is not measured on ships of opportunity?

**According to numerous previous studies and our own sensitivity test presented in Appendix C1, changes in $C_T$\* can be calculated from $pCO_2$ without exact knowledge of $A_T$, i.e. the bias in $\Delta C_T$\* is about 1 µmol kg$^{-1}$ for a bias in $A_T$ of about 10 µmol kg$^{-1}$ (Fig. C1). For the Baltic Sea, a rough $A_T$ estimate based on the well known and frequently monitored $A_T$-S relationship is sufficient to derive $C_T$\* with acceptable uncertainty, i.e. with a conversion uncertainty that is much lower than other sources of uncertainty, such as regional variability. We conclude that the requirement of a mean $A_T$ estimate does not weaken the utility of our approach, except for regions where $A_T$ is very poorly constrained. We clarified this in the main text by adding the following text in l.501:**

*Likewise, the required mean $A_T$ estimate should not restrict the applicability of our approach even if $A_T$ is not directly measured. For the Baltic Sea, it was demonstrated (Schneider et al., 2003) that $A_T$ estimated from the known $A_T$–S relationship (Müller et al., 2016) is sufficiently accurate to convert $pCO_2$ to $C_T$\* (see also Appendix C1).*

**Additional figures for review purpose only**

[Figure]

**Fig. 4 as in the manuscript, but without data from stations 07-10.**

[Figure]

**Fig. 5 as in the manuscript, but without data from stations 07-10.**